# Poison as Cure: Visual Noise for Mitigating Object Hallucinations in LVMs

**Kejia Zhang**[1]    **Keda Tao**[2]    **Jiasheng Tang**[3,4]    **Huan Wang**[2]*
[1]Xiamen University    [2]Westlake University
[3]DAMO Academy, Alibaba Group    [4]Hupan Lab
Project Page: https://kejiazhang-robust.github.io/poison-cure-lvm

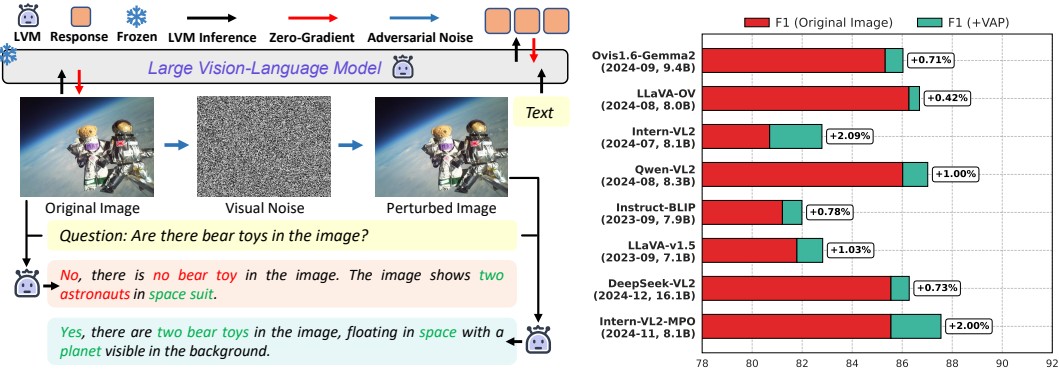

Figure 1: **Left:** We introduce *VAP* (visual adversarial perturbation), a training-free approach that strategically injects beneficial visual noise to mitigate object hallucination in LVMs without altering the complex base model. **Right:** Our method consistently improves performance across 8 state-of-the-art LVMs under the POPE hallucination evaluation setting [26].

## Abstract

Large vision-language models (LVMs) extend large language models (LLMs) with visual perception capabilities, enabling them to process and interpret visual information. A major challenge compromising their reliability is object hallucination that LVMs may generate plausible but factually inaccurate information. We propose a novel *visual adversarial perturbation (VAP)* method to mitigate this hallucination issue. VAP alleviates LVM hallucination by applying strategically optimized visual noise without altering the base model. Our approach formulates hallucination suppression as an optimization problem, leveraging adversarial strategies to generate beneficial visual perturbations that enhance the model's factual grounding and reduce parametric knowledge bias. Extensive experimental results demonstrate that our method consistently reduces object hallucinations across 8 state-of-the-art LVMs, validating its efficacy across diverse evaluations.

## 1 Introduction

Large vision-language models (LVMs) integrate visual and textual information, providing capabilities for addressing complex cross-modal understanding challenges [43, 5, 38, 22]. Despite their remarkable advancements, LVMs often generate plausible yet factually inaccurate outputs, eliciting harmful content such as misinformation or biased representations [26, 33]. Addressing these limitations is critical to enhancing the reliability and applicability of LVMs in real-world scenarios.

---

*Corresponding author.

39th Conference on Neural Information Processing Systems (NeurIPS 2025).

Prior research indicates that hallucinations in LVMs arise from the interaction between biased parametric knowledge and real-world data distributions [2, 15, 11]. This phenomenon is driven by two primary mechanisms. First, the long-tail distribution of training data induces systematic biases in parametric knowledge, resulting in spurious correlations and factual inconsistencies [26, 27]. Second, the extensive parameter spaces of large language models (LLMs) within LVMs amplify these biases, particularly given the LLMs' predominant role in the inference pipeline [23, 30, 39]. This LLM dominance potentially suppresses critical visual signals, increasing hallucination frequency [36, 42, 24]. Consequently, the embedded biased parametric knowledge substantially compromises LVMs' capacity to accurately process real-world data.

Existing solutions mitigate this challenge via two strategies: fine-tuning [27, 54, 4] and decoder optimization [17, 30, 8]. These model-centric interventions adjust LVMs' internal mechanisms through parametric updates or algorithmic refinements [29]. They have achieved substantial success in reducing hallucinations, laying crucial groundwork for improving LVM reliability.

Unlike prior model-centric approaches, we introduce a paradigm shift in hallucination mitigation that leverages the intrinsic mechanisms of hallucinations. This perspective stems from a crucial observation that while hallucinations arise from biased parametric knowledge, they manifest specifically during the processing of real-world visual inputs [16, 2]. This understanding reveals an elegant solution: strategically crafted perturbations to visual inputs can redirect LVMs' decision-making processes away from parametric biases without altering the original model's architecture or mechanisms.

This insight motivates our visual adversarial perturbation strategy, where adversarial optimization through zero-gradient techniques introduces beneficial visual noise to the original image. This noise guides the model to ground its responses in actual visual content rather than relying on parametric knowledge biases. The power of this approach lies in its exploitation of visual inputs as concrete factual anchors, fundamentally different from language prompts that often reinforce existing parametric biases [41, 50]. Notably, our method functions in a fully black-box manner requiring no access or modification to the LVM, making it a practical and efficient solution.

Building on this foundation, we propose visual adversarial perturbation (VAP), a novel technique designed to mitigates hallucinations by applying beneficial adversarial perturbations to visual inputs (as shown in Figure 1 (left)). Adversarial perturbations, traditionally considered as "poison" due to their initial disruption of model decisions, are reformulated to specifically align model responses with visual content and mitigate parametric knowledge bias. By adversarially optimizing visual noise, VAP refines LVM decision-making in a data-centric manner, transforming perturbations from a factor of degradation into a corrective "cure" that effectively mitigates object hallucinations.

We evaluate the effectiveness of VAP using complementary hallucination assessment frameworks: POPE [26] and BEAF [53] for closed VQA evaluation, and CHAIR [36] for open-ended generation tasks. Our extensive experiments across 8 state-of-the-art (SOTA) LVMs demonstrate that VAP consistently mitigates hallucinations across diverse evaluation settings.

Overall, our contributions are structured as follows:

- We propose visual adversarial perturbation, which mitigates object hallucinations in LVMs by injecting beneficial adversarial noise into visual inputs without modifying the model.

- We formulate object hallucination mitigation as an adversarial visual noise optimization. By refining adversarial strategies, beneficial visual noise is generated through zero-gradient optimization to influence model decision-making and alleviate hallucinations.

- Extensive experiments across evaluation settings—including text-axis, text- and vision-axes, and open-ended captioning—validate the efficacy of our method in reducing hallucinations.

## 2 Related Work

### 2.1 Large-Vision Language Models

In recent years, the field has witnessed advancements in large vision-language models (LVMs). LVMs have been developed to tackle real-world multimodal challenges such as image captioning and visual question answering [52, 47, 40, 58]. They typically operate through a pipeline comprising a visual encoder, a cross-modal connector, and a large language model (LLM), enabling seamless interaction

between visual and linguistic features. State-of-the-art systems leverage extensive datasets and adopt a two-stage training paradigm: pretraining on diverse multimodal corpora [35, 37], followed by fine-tuning with task-specific instructions [28, 32]. This methodology allows LVMs to interpret and respond to complex multimodal inputs with remarkable efficacy [25, 10].

## 2.2 Hallucination in LVMs

Hallucination refers to the generation of textual responses that deviate from or contradict the actual visual content, leading to factual inaccuracies or biased information in LVMs [26, 3, 2]. These hallucinations primarily arise from intrinsic limitations of LVMs, specifically: (1) the long-tail distribution of training data, which introduces systematic biases into the model's parametric knowledge [57, 54]; and (2) the vast parameter space of LLMs, which dominate the inference process and exacerbate these biases [29, 30]. Due to the fundamental role of objects in computer vision and multimodal research, current evaluation frameworks primarily concentrate on object hallucination [36, 57].

Prior work has explored two model-centric strategies to mitigate object hallucinations in LVMs: fine-tuning and decoding strategies. These interventions target the underlying parametric knowledge bias that leads to hallucinations. Fine-tuning approaches like REVERIE [21] and HalluciDoctor [54] update the parametric knowledge through comprehensive instruction data to suppress hallucinations. Meanwhile, decoding-based methods such as PDM [12] and OPERA [17] mitigate hallucinations by intervening in the model's decoding process. In contrast to these model-centric strategies, we approach the challenge from a data-centric perspective, proposing a novel adversarial visual perturbation technique that directly mitigates object hallucinations through visual perturbations.

## 3 Methodology

We propose visual adversarial perturbation (VAP) to mitigate object hallucination in LVMs. VAP formulates an adversarial strategy to align the LVM responses with visual content while reducing the impact of parametric knowledge bias (Section 3.2). These objectives guide the adversarial optimization process, which generates beneficial visual noise to improve model performance (Section 3.3). An overview of our framework is shown in Figure 2.

### 3.1 Preliminaries

**Notations** Let $f_\theta$ denote LVM, where $x$ represents the input image, $c$ is the query prompt, and $w$ is the model's generated response, such that $w = f_\theta(x, c)$. We define $g_\psi$ as the CLIP text encoder converting textual data into semantically meaningful embeddings. For adversarial perturbation, we denote $\delta$ as the perturbation vector and $\mathcal{L}_S$ as the surrogate adversarial loss guided by strategy set $S = [s_1, \cdots, s_n]$. The perturbed image is defined as $\hat{x} = x + \delta$, $\epsilon$ is the magnitude of perturbation, and $\Omega$ represents the adversarial knowledge utilized during the adversarial optimization process.

**Adversarial Perturbation** Adversarial perturbation against LVMs typically involves adding imperceptible visual noise to influence model decisions [56, 9], which can significantly alter the model's output. The optimization of such perturbations can be formulated as:

$$\delta = \arg\max_{\delta \sim \mathbb{B}_\epsilon(x)} \mathcal{L}_{(S)}(x + \delta, \Omega), \tag{1}$$

where $\delta$ represents the adversarial perturbation to be optimized, $\mathcal{L}_{(S)}$ represents the adversarial objective function under strategy $S$, and $\Omega$ indicates the available adversarial knowledge. The perturbation is bounded within an $\epsilon$-ball $\mathbb{B}$. Specifically, the adversarial perturbation is optimized by computing the gradient as follows:

$$\hat{x} = x + \alpha \nabla_x \{\mathcal{L}_{(S)}(x + \delta, \Omega)\}, \tag{2}$$

where $\alpha$ is the step size, and the gradient $\nabla_x$ is computed with respect to the vision input $x$.

### 3.2 Adversarial Strategies

Our adversarial goal is formulated as two principal objectives: **(1)** optimizing the semantic alignment between the response and the corresponding visual content of LVMs, and **(2)** mitigating the negative influence of parametric knowledge bias.

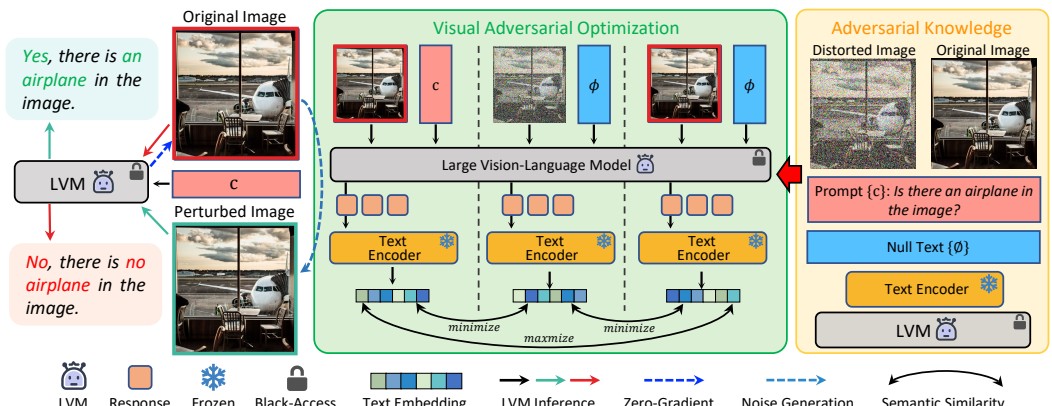

Figure 2: **Overview of our proposed method.** VAP generates visual noise by optimizing three strategies based on adversarial knowledge: (1) aligning responses under prompted and unprompted settings to preserve image-consistent semantics, (2) introducing uncertainty via distorted inputs to expose hallucination bias, and (3) minimizing representational similarity between original and distorted views to suppress parametric priors. Adversarial knowledge refers to structured conditions used to drive the optimization. The resulting perturbation mitigates object hallucinations.

**Alignment LVM Response with Grounding Visual Content** Hallucinations in LVMs manifest as the generation of semantically plausible responses but diverge from the actual visual content. To mitigate this, our proposed methodology promotes enhanced alignment between the model's responses and the actual visual content:

$$\mathcal{L}_{s_1} = \max_{\delta \sim \mathbb{B}_\epsilon(x)} \{ S(f_\theta(x + \delta, c), f_\theta(x + \delta, \emptyset)) \}, \tag{3}$$

where $S(\cdot, \cdot)$ signifies the calculation of semantic correlation between the two generated responses, $f_\theta(x + \delta, c)$ represents the model's output given the perturbed vision input $x + \delta$ with the conditional query prompt $c$, and $f_\theta(x + \delta, \emptyset)$ signifies the visual semantic description when the prompt is replaced with an empty token $\emptyset$. This loss term $\mathcal{L}_{s_1}$ quantifies the semantic alignment between conditionally guided responses and the model's autonomous interpretation of visual content, thereby enhancing response consistency with the underlying visual semantics.

Despite the improvements, the alignment between responses and visual content may still be influenced by parametric knowledge bias, particularly an over-reliance on linguistic priors [4]. Such bias can distort the model's interpretation of visual information, leading to hallucinatory patterns. As discussed in Section 1, LVMs often prioritize linguistically anchored priors over visual signals, thereby exacerbating existing biases. Our alignment strategy addresses this by mitigating both misalignment and bias.

**Mitigating Parametric Knowledge Bias** Visual uncertainty [15, 24, 51] serves as a critical metric for quantifying parametric knowledge bias. It is quantified by generating a contrastive negative image $\bar{x}$ through the introduction of noise to the original image:

$$p(\bar{x}|x) = \mathcal{N}(\bar{x}; \sqrt{\mu_T} x, (1 - \mu_T)\mathbf{I}), \tag{4}$$

where $\mu_T$ represents the noise scheduling coefficient at timestep $T$, controlling the magnitude of perturbation applied to the original image $x$.

To further mitigate parametric knowledge bias, we introduce a dual-setting approach that reduces the semantic similarity between LVM responses to original and distorted visual inputs under both conditional $c$ (with query prompt) and unconditional $\emptyset$ (without query prompt) configurations.

In the conditional $c$ setting, our approach minimizes the semantic similarity between the perturbed input $x + \delta$ and the contrastive negative image $\bar{x}$:

$$\mathcal{L}_{s_2} = \min_{\delta \sim \mathbb{B}_\epsilon(x)} \{ S(f_\theta(x + \delta, c), f_\theta(\bar{x}, \emptyset)) \}, \tag{5}$$

where $f_\theta(\bar{x}, \emptyset)$ denotes the LVM's output given the visually uncertain input. $\mathcal{L}_{s_2}$ promotes more discriminative and context-sensitive responses between prompted and unprompted conditions, thereby effectively reducing the model's dependency on linguistic priors.

In the unconditional $\emptyset$ setting, our methodology minimizes the semantic similarity between responses to the perturbed image $x + \delta$ and its contrastive negative counterpart $\bar{x}$:

$$\mathcal{L}_{s_3} = \min_{\delta \sim \mathbb{B}_\epsilon(x)} \{ S(f_\theta(x + \delta, \emptyset), f_\theta(\bar{x}, \emptyset)) \}, \tag{6}$$

where $\mathcal{L}_{s_3}$ alleviates the propensity to hallucinate, further mitigating the dominant influence of linguistic priors.

The loss terms $\mathcal{L}_{s_1}$, $\mathcal{L}_{s_2}$, and $\mathcal{L}_{s_3}$ collectively regulate LVM responses to ensure consistency with visual content while mitigating parametric knowledge bias in LVMs. We formulate our complete optimization objective as a weighted combination of these loss terms:

$$\mathcal{L}_S(x, c, \theta) = \frac{\mathcal{L}s_1}{\sigma_1^2} + \frac{\mathcal{L}s_2}{\sigma_2^2} + \frac{\mathcal{L}s_3}{\sigma_3^2}, \tag{7}$$

where $\sigma_i^2$ ($i \in \{1, 2, 3\}$) are balancing coefficients that modulate the contribution of each loss component. This formulation achieves a dual objective: $\mathcal{L}_{s_1}$ ensures strong semantic alignment between model responses and visual content, while $\mathcal{L}_{s_2}$ and $\mathcal{L}_{s_3}$ collectively mitigate parametric knowledge bias through consistent interpretation across visual perturbations.

### 3.3 Visual Adversarial Optimization

To optimize our adversarial objectives $\mathcal{L}_S$, we leverage the CLIP text encoder $g_\psi(\cdot)$ as a surrogate model, capitalizing on its superior discriminative capabilities for textual representation [48]. This approach contrasts with the limited semantic separability in LLM representations:

$$S(\cdot, \cdot) = g_\psi(\cdot)^\top g_\psi(\cdot), \tag{8}$$

where $S(\cdot, \cdot)$ measures the similarity of the LVM's response under different conditions. Then, we compute the numerical loss $\mathcal{L}_S(x, c, \theta)$, which enables the optimization of the perturbation $\delta$. $\delta$ represents a carefully crafted visual perturbation designed to optimize the strategic objective:

$$\delta = \nabla_x \{ \mathcal{L}_S(x, c, \theta, \psi) \}. \tag{9}$$

The final adversarial perturbation is generated by adding noise to the input image $x$, yielding the visual adversarial perturbed image $\hat{x}$:

$$\hat{x} = x + \alpha \cdot \delta = x + \alpha \nabla_x \{ \mathcal{L}_S(x, c, \theta, \psi) \}, \tag{10}$$

where $\alpha$ denotes the learning rate of adversarial strategies. The generated perturbed image $\hat{x}$ exhibits superior optimization characteristics with respect to the objective $\mathcal{L}_S$, outperforming the original images $x$ while meticulously preserving the semantic integrity of vision input.

Due to the autoregressive nature of LVMs, direct gradient computation is challenging. To address this, we optimize the similarity-based loss using a gradient-free method [56, 34], termed zero-gradient optimization. Specifically, we apply a zero-order optimization technique [6], which approximates the gradient by evaluating the loss at perturbed inputs and estimating the optimal perturbation direction:

$$\nabla_x \{ \mathcal{L}_S(x, c, \theta) \} \approx \frac{1}{N \cdot \beta} \sum_{n=1}^N \{ [\mathcal{L}_S(x + \beta \cdot \gamma_n, c, \theta, \psi) \\ - \mathcal{L}_S(x, c, \theta, \psi)] \cdot \gamma_n \}, \tag{11}$$

where $\gamma_n$ is sampled from distribution $P(\gamma)$, $\beta$ controls the sampling variance, and $N$ denotes the number of queries. The term $\gamma_n \sim P(\gamma)$ ensures perturbation diversity through the property $E[\gamma^\top \cdot \gamma] = I$. A detailed step-by-step algorithm of VAP is provided in Appendix I.

## 4 Experiments

To thoroughly assess VAP, we conduct experiments from five perspectives:

- **Consistency:** Evaluating VAP's effectiveness in mitigating hallucinations across eight LVMs.
- **Fidelity:** Ensuring that visual understanding and reasoning capabilities are preserved.
- **Compatibility:** Demonstrating VAP's orthogonality to other methods and complementary benefits.
- **Efficiency:** Reducing computational cost via a lightweight solution achieving $1/8\times$ overhead.
- **Component Analysis:** Assessing the contribution of each module through ablation.

## 4.1 Experiment Setup

**Implementation Details** We evaluated our method on 8 SOTA LVMs: LLaVA [28], LLaVA-Onevision (OV) [25], Instruct-BLIP [10], Intern-VL2 [7], Intern-VL2-MPO [7], Qwen-VL2 [46], DeepSeek-VL2 [49], and Ovis1.6-Gemma2 [31]. In our experiments, we set the parameters as $\alpha = 1/255$, $\beta = 8/255$, $N = 10$, and $\epsilon = 2$. Due to the differences across LVMs, we assigned model-specific balancing coefficients $\sigma_i$ (where $i \in 1, 2, 3$) and $T$.

Detailed model descriptions, configurations are provided in Appendix A. Additionally, an in-depth analysis of the ablation study and individual components can be found in Appendix C.

**Evaluation Benchmark** Our evaluation is divided into two main categories: **(1) Closed VQA for object hallucination evaluation:** Text-axis evaluation POPE [26] and vision-/text-axis evaluation BEAF [53] settings. **(2) Open-ended evaluation:** Image caption generation CHAIR [36] setting. **(3) Non-hallucination evaluation:** Factual object recognition and open-ended factual understanding tasks using MME [14] and AMBER [45] (See in Appendix J.1). Further details are provided in Appendix B, and comprehensive examples are presented in Appendix E.

**1) POPE:** POPE evaluates hallucinations along the text axis by generating VQA pairs through question manipulation. We randomly selected 500 samples from the MS-COCO dataset and generated 9,000 evaluation triplets using POPE's three sampling strategies.

**2) BEAF:** BEAF evaluates hallucinations along vision/text axes by manipulating scene information and questions for fine-grained analysis. BEAF incorporates change-aware metrics such as TU, IG, $SB_p$, $SB_n$, ID, and $F1_{TUID}$ for comprehensive evaluation. BEAF includes 26,064 evaluation triplets.

**3) CHAIR:** CHAIR evaluates hallucination by generating captions and measuring the proportion of objects mentioned in captions but not present in images. Specifically, we randomly select 1,000 samples from the MS-COCO dataset for evaluation. The assessment uses two metrics:

$$\text{CHAIR}_I = \frac{|\text{hallucinated objects}|}{|\text{captioned objects}|}, \quad \text{CHAIR}_S = \frac{|\text{hallucinated captions}|}{|\text{all captions}|} \tag{12}$$

where $\text{CHAIR}_I$ is calculated at the object level, and $\text{CHAIR}_S$ is calculated at the sentence level.

**4) AMBER/MME:** AMBER and MME serve as comprehensive evaluation benchmarks for multimodal large language models. They assess various attributes of multimodal capabilities, focusing on both perception and cognition in discriminative and generative tasks.

## 4.2 Experimental Results

**Results on text-axis hallucination evaluation** Table 1 presents comparative results under the POPE (Polling-based Object Probing Evaluation) setting[2]. Our experimental methodology includes three sampling strategies: Random, Popular, and Adversarial Sampling for negative object selection, each generating 3,000 evaluation triplets. Across all settings, integrating VAP through visual noise injection consistently improved the performance of eight state-of-the-art LVMs, with the largest gains observed in Intern-VL2: +2.81% in accuracy and +2.09% in F1 score. Notably, the most significant improvements appear under adversarial sampling (Figure 1-right), indicating that VAP effectively mitigates parametric knowledge bias in LVMs. This is particularly relevant as adversarial sampling tends to trigger high-frequency hallucinated objects, highlighting the data distribution bias in LVM training and the dominant role of LLMs.

**Results on Vision-/Text-Axis Hallucination Evaluation** Table 2 presents comparative results under the BEAF (BEfore-AFter) framework, which enables fine-grained analysis through vision-axis manipulation and change-aware metrics, offering deeper insight than standard accuracy. Applying VAP led to consistent improvements across most metrics for all LVMs.

Notably, TU improved by 2.31%, $SB_p$ by 1.76%, $SB_n$ by 1.04%, and $F1_{TUID}$ by 1.74%. Gains across TU, IG, $SB_p$, $SB_n$, ID, and $F1_{TUID}$ indicate that VAP mitigates hallucinations under varied scene conditions by promoting genuine object understanding over spurious correlations. The marked TU gains further suggest that VAP's visual perturbations guide models toward more grounded predictions, validating its role in suppressing parametric bias and enhancing visual reasoning [53].

---

[2]Due to space limitations, complete precision and recall results are provided in Appendix C.1.

Table 1: Text-axis evaluation comparison under three evaluation settings of POPE on the validation set of MSCOCO: Random Sampling (selecting absent objects), Popular Sampling (choosing the most frequent missing objects based on dataset-wide occurrence), and Adversarial Sampling (ranking objects by co-occurrence with ground-truth and selecting the most frequent ones). The values in green indicate the percentage improvements achieved by our proposed method.

| LVM | Vision Input | Popular | | Random | | Adversarial | |
|---|---|---|---|---|---|---|---|
| | | Acc.↑ | F1↑ | Acc.↑ | F1↑ | Acc.↑ | F1↑ |
| LLaVA-v1.5 | *Original* | 85.57 | 86.19 | 88.97 | 89.09 | 79.80 | 81.79 |
| | +*VAP* | **86.67** $^{+1.10}$ | **87.18** $^{+0.99}$ | **90.00** $^{+1.03}$ | **90.07** $^{+0.98}$ | **80.97** $^{+1.17}$ | **82.82** $^{+1.03}$ |
| Instruct-BLIP | *Original* | 83.30 | 82.85 | 88.13 | 87.18 | 81.33 | 81.21 |
| | +*VAP* | **84.06** $^{+0.76}$ | **83.67** $^{+0.82}$ | **89.00** $^{+0.87}$ | **88.12** $^{+0.99}$ | **82.03** $^{+0.70}$ | **81.99** $^{+0.78}$ |
| Intern-VL2 | *Original* | 84.11 | 81.64 | 85.14 | 82.60 | 82.00 | 80.70 |
| | +*VAP* | **86.18** $^{+2.07}$ | **84.19** $^{+2.00}$ | **86.30** $^{+1.16}$ | **84.08** $^{+1.48}$ | **84.81** $^{+2.81}$ | **82.79** $^{+2.09}$ |
| Intern-VL2-MPO | *Original* | 87.51 | 86.53 | 88.68 | 87.58 | 86.28 | 85.55 |
| | +*VAP* | **89.08** $^{+1.57}$ | **88.27** $^{+1.74}$ | **90.20** $^{+1.52}$ | **89.30** $^{+1.72}$ | **88.13** $^{+1.85}$ | **87.55** $^{+2.00}$ |
| DeepSeek-VL2 | *Original* | 86.80 | 85.86 | 88.70 | 87.64 | 86.47 | 85.55 |
| | +*VAP* | **87.60** $^{+0.80}$ | **86.70** $^{+0.84}$ | **89.30** $^{+0.60}$ | **88.31** $^{+0.67}$ | **87.13** $^{+0.66}$ | **86.28** $^{+0.73}$ |
| Qwen-VL2 | *Original* | 88.13 | 87.68 | 90.60 | 89.99 | 86.27 | 86.02 |
| | +*VAP* | **89.10** $^{+0.97}$ | **88.65** $^{+0.97}$ | **91.16** $^{+0.56}$ | **90.54** $^{+0.55}$ | **87.30** $^{+1.03}$ | **87.02** $^{+1.00}$ |
| LLaVA-OV | *Original* | 88.30 | 87.33 | 89.53 | 88.51 | 87.17 | 86.27 |
| | +*VAP* | **88.93** $^{+0.63}$ | **87.93** $^{+0.60}$ | **89.87** $^{+0.34}$ | **88.83** $^{+0.32}$ | **87.76** $^{+0.59}$ | **86.69** $^{+0.42}$ |
| Ovis1.6-Gemma2 | *Original* | 87.96 | 86.88 | 88.96 | 87.87 | 86.22 | 85.32 |
| | +*VAP* | **88.44** $^{+0.48}$ | **87.40** $^{+0.52}$ | **89.59** $^{+0.65}$ | **88.54** $^{+0.67}$ | **86.85** $^{+0.63}$ | **86.03** $^{+0.71}$ |

Table 2: Vision-/text-Axis evaluation comparison under the BEAF Benchmark. Compared to the text-axis hallucination evaluation, BEAF includes the change-aware hallucination metrics: TU, IG, $SB_p$, $SB_n$, ID, and $F1_{TUID}$. Although some metrics show slight degradation, the overall performance demonstrates consistent improvement. The values in green indicate the percentage improvements achieved by our proposed method, while the values in red reflect the performance degradation.

| LVM | Vision Input | BEAF Benchmark | | | | | | | |
|---|---|---|---|---|---|---|---|---|---|
| | | Acc.↑ | F1↑ | TU↑ | IG↓ | $SB_p$↓ | $SB_n$↓ | ID↓ | $F1_{TUID}$↑ |
| LLaVA-v1.5 | *Original* | 79.99 | 74.06 | 34.25 | 0.33 | 60.74 | 4.66 | 5.42 | 50.31 |
| | +*VAP* | **80.36** $^{+0.37}$ | **74.35** $^{+0.29}$ | **34.83** $^{+0.58}$ | **0.27** $^{-0.06}$ | **60.72** $^{-0.02}$ | **4.18** $^{-0.46}$ | **5.05** $^{-0.37}$ | **50.97** $^{+0.66}$ |
| Instruct-BLIP | *Original* | 81.91 | 73.55 | 33.35 | 0.78 | 50.73 | 15.12 | 5.45 | 49.30 |
| | +*VAP* | **82.07** $^{+0.16}$ | **73.96** $^{+0.41}$ | **33.83** $^{+0.48}$ | **0.48** $^{-0.30}$ | **50.59** $^{-0.14}$ | **15.10** $^{-0.02}$ | **5.30** $^{-0.15}$ | **49.85** $^{+0.55}$ |
| Intern-VL2 | *Original* | 88.38 | 79.10 | 64.12 | 1.33 | 12.63 | 21.89 | 6.20 | 76.17 |
| | +*VAP* | **88.69** $^{+0.31}$ | **79.72** $^{+0.62}$ | **66.15** $^{+2.03}$ | **0.97** $^{-0.36}$ | **11.58** $^{-1.05}$ | **21.28** $^{-0.61}$ | **6.05** $^{-0.15}$ | **77.63** $^{+1.46}$ |
| Intern-VL2-MPO | *Original* | 89.21 | 82.56 | 63.24 | 0.76 | 23.67 | 12.31 | 5.23 | 75.86 |
| | +*VAP* | **89.63** $^{+0.42}$ | **82.72** $^{+0.18}$ | **65.06** $^{+1.78}$ | **0.45** $^{-0.31}$ | **21.91** $^{-1.76}$ | **12.55** $^{+0.24}$ | **4.49** $^{-0.74}$ | **77.40** $^{+1.66}$ |
| DeepSeek-VL2 | *Original* | 89.39 | 82.51 | 67.04 | 0.50 | 17.88 | 14.56 | 3.02 | 79.27 |
| | +*VAP* | **89.72** $^{+0.33}$ | **83.12** $^{+0.61}$ | **68.11** $^{+1.07}$ | **0.44** $^{-0.06}$ | **17.37** $^{-0.51}$ | **14.06** $^{-0.50}$ | **2.98** $^{-0.04}$ | **80.03** $^{+0.76}$ |
| Qwen-VL2 | *Original* | 87.96 | 81.13 | 54.78 | 0.28 | 33.68 | 11.24 | 4.89 | 69.78 |
| | +*VAP* | **88.39** $^{+0.43}$ | **81.57** $^{+0.44}$ | **56.18** $^{+1.40}$ | **0.27** $^{-0.01}$ | **32.49** $^{-1.19}$ | **11.03** $^{-0.21}$ | **4.38** $^{-0.51}$ | **70.79** $^{+1.01}$ |
| LLaVA-OV | *Original* | 90.76 | 84.53 | 65.80 | 0.12 | 21.32 | 12.77 | 2.55 | 78.56 |
| | +*VAP* | **91.07** $^{+0.33}$ | **85.01** $^{+0.48}$ | **67.16** $^{+1.36}$ | **0.30** $^{+0.18}$ | **20.81** $^{-0.51}$ | **11.73** $^{-1.04}$ | **2.46** $^{-0.09}$ | **79.54** $^{+0.98}$ |
| Ovis1.6-Gemma2 | *Original* | 90.12 | 83.04 | 66.25 | 0.28 | 19.94 | 13.52 | 2.76 | 78.80 |
| | +*VAP* | **90.91** $^{+0.79}$ | **84.53** $^{+1.49}$ | **68.56** $^{+2.31}$ | **0.25** $^{-0.03}$ | **19.69** $^{-0.25}$ | **11.48** $^{-2.04}$ | **2.41** $^{-0.25}$ | **80.54** $^{+1.74}$ |

**Results on Open-Ended Caption Generation Hallucination Evaluation** Table 4 reports our model's performance under the CHAIR (Caption Hallucination Assessment with Image Relevance) setting.[3] Applying optimized VAP to original images yields consistent reductions in object hallucination across diverse query prompts. For example, under the prompt "Generate a short caption of the image," Intern-VL2 achieves $CHAIR_I$ and $CHAIR_S$ reductions of 0.68 and 0.90, respectively, with VAP.

These results highlight VAP's effectiveness in open-ended vision-language tasks beyond binary VQA. By mitigating hallucination, VAP improves the semantic alignment between captions and visual content, reduces parametric bias, and enhances the factuality and relevance of generated descriptions.

---

[3]CHAIR is limited to 80 segmentation categories, which may induce classification bias [26]. We restrict responses to 30 characters to focus on prominent objects.

Table 3: Comparison of VAP improvements on POPE (text-axis) and BEAF (vision/text-axis). Only Accuracy and F1 are shown for compactness. Green numbers indicate performance gains.

| LVM | Vision Input | POPE-Popular | | POPE-Random | | POPE-Adversarial | | BEAF | |
|---|---|---|---|---|---|---|---|---|---|
| | | Acc↑ | F1↑ | Acc↑ | F1↑ | Acc↑ | F1↑ | Acc↑ | F1↑ |
| LLaVA-v1.5 | Original | 85.57 | 86.19 | 88.97 | 89.09 | 79.80 | 81.79 | 79.99 | 74.06 |
| | +VAP | $86.67^{+1.10}$ | $87.18^{+0.99}$ | $90.00^{+1.03}$ | $90.07^{+0.98}$ | $80.97^{+1.17}$ | $82.82^{+1.03}$ | $80.36^{+0.37}$ | $74.35^{+0.29}$ |
| Instruct-BLIP | Original | 83.30 | 82.85 | 88.13 | 87.18 | 81.33 | 81.21 | 81.91 | 73.55 |
| | +VAP | $84.06^{+0.76}$ | $83.67^{+0.82}$ | $89.00^{+0.87}$ | $88.12^{+0.99}$ | $82.03^{+0.70}$ | $81.99^{+0.78}$ | $82.07^{+0.16}$ | $73.96^{+0.41}$ |
| Intern-VL2 | Original | 84.11 | 81.64 | 85.14 | 82.60 | 82.00 | 80.70 | 88.38 | 79.10 |
| | +VAP | $86.18^{+2.07}$ | $84.19^{+2.00}$ | $86.30^{+1.16}$ | $84.08^{+1.48}$ | $84.81^{+2.81}$ | $82.79^{+2.09}$ | $88.69^{+0.31}$ | $79.72^{+0.62}$ |
| Intern-VL2-MPO | Original | 87.51 | 86.53 | 88.68 | 87.58 | 86.28 | 85.55 | 89.21 | 82.56 |
| | +VAP | $89.08^{+1.57}$ | $88.27^{+1.74}$ | $90.20^{+1.52}$ | $89.30^{+1.72}$ | $88.13^{+1.85}$ | $87.55^{+2.00}$ | $89.63^{+0.42}$ | $82.72^{+0.18}$ |
| DeepSeek-VL2 | Original | 86.80 | 85.86 | 88.70 | 87.64 | 86.47 | 85.55 | 89.39 | 82.51 |
| | +VAP | $87.60^{+0.80}$ | $86.70^{+0.84}$ | $89.30^{+0.60}$ | $88.31^{+0.67}$ | $87.13^{+0.66}$ | $86.28^{+0.73}$ | $89.72^{+0.33}$ | $83.12^{+0.61}$ |
| Qwen-VL2 | Original | 88.13 | 87.68 | 90.60 | 89.99 | 86.27 | 86.02 | 87.96 | 81.13 |
| | +VAP | $89.10^{+0.97}$ | $88.65^{+0.97}$ | $91.16^{+0.56}$ | $90.54^{+0.55}$ | $87.30^{+1.03}$ | $87.02^{+1.00}$ | $88.39^{+0.43}$ | $81.57^{+0.44}$ |
| LLaVA-OV | Original | 88.30 | 87.33 | 89.53 | 88.51 | 87.17 | 86.27 | 90.76 | 84.53 |
| | +VAP | $88.93^{+0.63}$ | $87.93^{+0.60}$ | $89.87^{+0.34}$ | $88.83^{+0.32}$ | $87.76^{+0.59}$ | $86.69^{+0.42}$ | $91.07^{+0.33}$ | $85.01^{+0.48}$ |
| Ovis1.6-Gemma2 | Original | 87.96 | 86.88 | 88.96 | 87.87 | 86.22 | 85.32 | 90.12 | 83.04 |
| | +VAP | $88.44^{+0.48}$ | $87.40^{+0.52}$ | $89.59^{+0.65}$ | $88.54^{+0.67}$ | $86.85^{+0.63}$ | $86.03^{+0.71}$ | $90.91^{+0.79}$ | $84.53^{+1.49}$ |

Table 4: Comparison of object hallucination evaluation under the CHAIR setting. $I_1$ denotes "*Generate a short caption of the image*", and $I_2$ denotes "*Provide a brief description of the given image*". The values in green indicate the percentage improvements achieved by our proposed method.

| LVM | Vision Input | $I_1$ | | $I_2$ | |
|---|---|---|---|---|---|
| | | $\text{CHAIR}_I \downarrow$ | $\text{CHAIR}_S \downarrow$ | $\text{CHAIR}_I \downarrow$ | $\text{CHAIR}_S \downarrow$ |
| LLaVA-v1.5 | *Original* | 3.97 | 6.60 | 4.01 | 6.90 |
| | *+VAP* | $3.82^{-0.15}$ | $6.50^{-0.10}$ | $3.86^{-0.15}$ | $6.50^{-0.40}$ |
| Instruct-BLIP | *Original* | 1.83 | 2.90 | 2.14 | 3.40 |
| | *+VAP* | $1.71^{-0.12}$ | $2.70^{-0.20}$ | $1.96^{-0.18}$ | $3.10^{-0.30}$ |
| Intern-VL2 | *Original* | 4.90 | 7.50 | 5.14 | 9.50 |
| | *+VAP* | $4.22^{-0.68}$ | $6.60^{-0.90}$ | $4.65^{-0.49}$ | $8.90^{-0.60}$ |
| Intern-VL2-MPO | *Original* | 5.53 | 8.90 | 6.35 | 13.40 |
| | *+VAP* | $5.39^{-0.14}$ | $8.60^{-0.30}$ | $6.17^{-0.18}$ | $12.60^{-0.80}$ |
| DeepSeek-VL2 | *Original* | 2.00 | 2.60 | 1.84 | 4.50 |
| | *+VAP* | $1.94^{-0.06}$ | $2.20^{-0.40}$ | $1.66^{-0.18}$ | $4.30^{-0.20}$ |
| Qwen-VL2 | *Original* | 3.27 | 5.20 | 3.45 | 6.20 |
| | *+VAP* | $2.98^{-0.29}$ | $4.80^{-0.40}$ | $3.23^{-0.22}$ | $5.70^{-0.50}$ |
| LLaVA-OV | *Original* | 1.96 | 3.30 | 2.71 | 4.50 |
| | *+VAP* | $1.85^{-0.11}$ | $3.10^{-0.20}$ | $2.41^{-0.30}$ | $4.20^{-0.30}$ |
| Ovis1.6-Gemma2 | *Original* | 4.07 | 6.30 | 5.80 | 14.50 |
| | *+VAP* | $3.90^{-0.17}$ | $6.20^{-0.10}$ | $5.56^{-0.24}$ | $14.30^{-0.20}$ |

## 4.3 Analysis and Discussion

**Effectiveness of VAP and Gaussian noise on hallucinations** Figure 3 compares the impact of VAP and Gaussian noise applied to original images under equal-strength perturbations. Gaussian noise consistently degrades performance across eight models, while VAP preserves or improves it. This highlights VAP's effectiveness in three aspects: Firstly, VAP introduces beneficial semantic noise, whereas Gaussian noise increases uncertainty and disrupts visual features. Secondly, VAP enhances alignment between model outputs and visual content via its adversarial strategy, reducing hallucinations. Thirdly, unlike Gaussian noise, which merely blurs input, VAP semantically challenges the model to mitigate parametric knowledge bias.

**Illustration of the effectiveness on closed VQA and open-ended tasks** Figure 4 presents results from examples in closed vision-question-answer (VQA) and open-ended image captioning tasks. Panels (a) and (b) demonstrate that the visual noise introduced by our method suppresses object hallucinations in LVMs under scene-change situations without disrupting their normal perceptual capabilities (i.e., the noise does not lead to incorrect decisions). Additionally, Panels (c) and (d) show that our method mitigates object hallucinations in open-ended tasks without reducing the amount

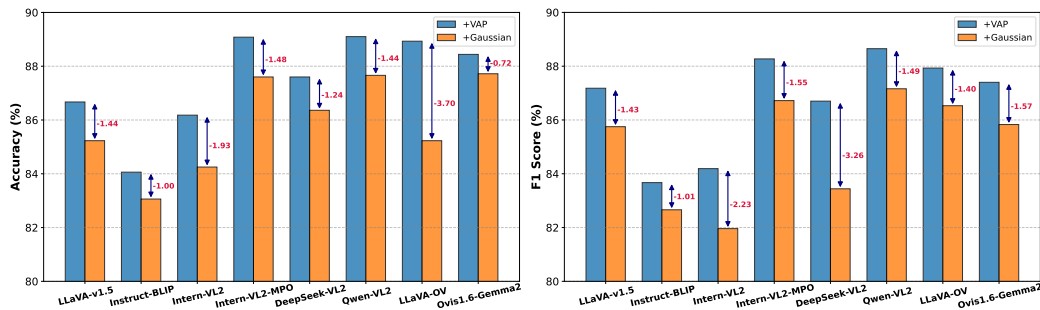

Figure 3: Comparison of original images with our VAP and Gaussian noise of equal strength ($\epsilon = 2$). We highlight the performance drop caused by Gaussian noise compared to VAP. Experiments were conducted under the POPE adversarial setting, evaluated by Accuracy and F1 Score.

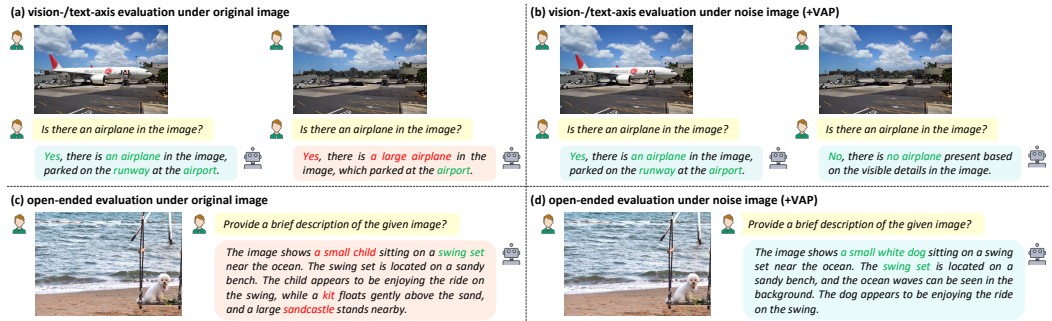

Figure 4: Examples of the vision-question-answer (VQA) tasks before and after applying our proposed method to the original images. (a) and (b) demonstrates the suppression of hallucinations in vision-/text-axis evaluations. (c) and (d) shows the reduction of hallucinations in open-ended tasks. Specifically, we use the LLaVA-v1.5 [28] as an example.

of information in the LVMs' responses. These consistent findings highlight the effectiveness of the VAP method. More comprehensive examples can be found in Appendix E. In-depth analyses of generalization are provided in Appendix D.

**Computational cost analysis and efficient proxy-based solution** We report on the computational cost of VAP optimization and present a more efficient approach. Our innovative proxy-based strategy leverages smaller-scale models to generate adversarial perturbations, which are then effectively transferred to larger models. As illustrated in Table 5, our approach reduces generation time by up to eightfold while maintaining comparable accuracy. Notably, VAP generated by the Intern-VL2-1B model and applied to the Intern-VL2-8B model achieves an accuracy of 84.07%, compared to 84.81% with self-generated VAP, with only a minor increase in runtime (+41ms vs. +298ms). This demonstrates that our proxy solution efficiently introduces beneficial noise that is generalizable across models, sustaining inference latency and enabling scalable deployment across large vision-language models, thus enhancing overall system efficiency.

## 5 Conclusion

This paper presents visual adversarial perturbation (VAP), an innovative data-centric, training-free method to reduce object hallucinations in large vision-language models (LVMs) by introducing imperceptible noise to visual inputs. Unlike model-centric approaches requiring complex modifications, VAP strategically applies beneficial noise to visual data, grounding model responses in actual content and reducing reliance on biased parametric knowledge. Extensive evaluations on the POPE, BEAF, CHAIR, AMBER, and MMH benchmarks show that VAP significantly decreases object hallucinations across various settings, enhancing LVM reliability.

Table 5: Computational cost and efficiency analysis of proxy-based VAP generation. The table presents the performance and runtime evaluation of Intern-VL2-8B [7] and Qwen-VL2-7B [46] under different vision input strategies. The proxy-based approach substantially reduces computational overhead while preserving strong hallucination suppression performance.

| LVM | Vision Input | Proxy Model | Accuracy(%) ↑ | Runtime (A100 per time) ↓ | Computational Cost ↓ |
|---|---|---|---|---|---|
| Intern-VL2-8B | *Original* | - | 82.00 | 160ms | - |
| | *+VAP* | Intern-VL2-8B | **84.81** (+2.81) | +298ms | 1× |
| | *+VAP-Proxy* | Intern-VL2-1B | 84.07 (+2.07) | **+39ms** | 1/8× |
| Qwen-VL2-7B | *Original* | - | 86.27 | 133ms | - |
| | *+VAP* | Qwen-VL2-7B | **87.30** (+1.03) | +245ms | 1× |
| | *+VAP-Proxy* | Qwen-VL2-2B | 86.87 (+0.60) | **+48ms** | 1/5× |

Our findings highlight the effectiveness of visual adversarial perturbations as a novel "poison as cure" strategy, uniquely demonstrated here. A key contribution is the consistent mitigation of model hallucinations in a black-box setting through noise addition, without compromising image understanding. Although VAP introduces computational overhead, we propose a proxy-based approach for efficient noise generation, maintaining performance while reducing costs to one-eighth. This work underscores VAP's potential as a transformative approach in enhancing LVM accuracy and reliability, paving the way for future research in data-centric model improvement.

## Acknowledgment

This paper is supported by Young Scientists Fund of the National Natural Science Foundation of China (No. 62506305), Zhejiang Leading Innovative and Entrepreneur Team Introduction Program (No. 2024R01007), Key Research and Development Program of Zhejiang Province (No. 2025C01026), Scientific Research Project of Westlake University (No. WU2025WF003), Chinese Association for Artificial Intelligence (CAAI) & Ant Group Research Fund - AGI Track (No. 2025CAAI-ANT-13), and the Special Support Talents Program of "Xi Hu Ming Zhu Program" in Hangzhou. We thank Xinjun Lin for the aesthetic insights provided in this paper.

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

## A More Details of Experiment Setup

### A.1 More Details about Baseline LVMs

In this study, we comprehensively selecte eight state-of-the-art large vision-language models (LVMs) carefully selected to validate the effectiveness of our proposed method. As illustrated in Table 6, our chosen models span critical developments from September 2023 to December 2024, encompassing parameter ranges from 7.1B to 16.1B and integrating advanced language models like Vicuna, Qwen2, and Gemma2 with sophisticated vision encoders such as CLIP, SigLIP, and custom vision transformers. Our model selection strategy focuses on capturing the latest architectural innovations in addressing hallucination challenges in vision-language understanding. By examining models from leading research initiatives including LLaVA, Instruct-BLIP, Intern-VL, DeepSeek, Ovis, LLaVA-OV and Qwen, we aim to provide a comprehensive hallucination evaluations of current multimodal AI.

Table 6: Detailed information of large vision-language models used in this paper.

| LVM | # Parameters | Language Model | Vision Model | Released Date |
|---|---|---|---|---|
| LLaVA-v1.5 [28] | 7.1B | Vicuna-7B | CLIP ViT-L/14 | 2023-09 |
| Instruct-BLIP [10] | 7.9B | Vicuna-7B | ViT-G | 2023-09 |
| Intern-VL2 [7] | 8.1B | InternLM2.5-7B | InternViT-300M | 2024-07 |
| Intern-VL2-MPO [7] | 8.1B | InternLM2.5-7B | InternViT-300M | 2024-11 |
| DeepSeek-VL2 [49] | 16.1B | DeepSeekMoE-16B | SigLIP-400M | 2024-12 |
| Qwen-VL2 [46] | 8.3B | Qwen2-7B | ViT-Qwen | 2024-08 |
| LLaVA-OV [25] | 8.0B | Qwen2-7B | SigLIP-400M | 2024-08 |
| Ovis1.6-Gemma2 [31] | 9.4B | Gemma2-9B | SigLIP-400M | 2024-11 |

### A.2 More Details about Implementation Details

We conducted our experiments across eight state-of-the-art vision-language models: LLaVA-v1.5, Instruct-BLIP, Intern-VL2, Intern-VL2-MPO, DeepSeek-VL2, Qwen-VL2, LLaVA-OV, and Ovis1.6-Gemma2. The experiments were performed using NVIDIA RTX 4090 (24GB), A6000 (48GB), and A100 (80GB) GPUs. For the adversarial parameters, we set $\alpha = 1/255$, $\beta = 8/255$, $N = 10$, and $\epsilon = 2$ unless otherwise noted. Model-specific balance parameters are detailed in Table 7. We employ ViT-L/14 as our default CLIP text encoder ($g_\psi$) unless otherwise specified.

Table 7: Detailed specifications of large vision-language models used in this paper.

| LVM | $\sqrt{1/\sigma_1{}^2}$ | $\sqrt{1/\sigma_2{}^2}$ | $\sqrt{1/\sigma_3{}^2}$ | $T$ |
|---|---|---|---|---|
| LLaVA-v1.5 [28] | 1.0 | 1.0 | 1.0 | 500 |
| Instruct-BLIP [10] | 1.0 | 1.0 | 1.0 | 500 |
| Intern-VL2 [7] | 1.0 | 0.5 | 0.5 | 200 |
| Intern-VL2-MPO [7] | 1.0 | 0.5 | 0.5 | 800 |
| DeepSeek-VL2 [49] | 1.0 | 1.0 | 1.0 | 100 |
| Qwen-VL2 [46] | 1.0 | 0.5 | 0.5 | 500 |
| LLaVA-OV [25] | 0.1 | 1.0 | 0.1 | 200 |
| Ovis1.6-Gemma2 [31] | 1.0 | 1.0 | 1.0 | 500 |

## B More Details of Evaluation Benchmark

### B.1 POPE Evaluation

POPE (Polling-based Object Probing Evaluation) [26] is a simple yet effective framework for assessing object hallucinations in LVMs. POPE formulates the evaluation of object hallucinations as a series of binary (yes/no) classification tasks. By sampling hallucinated objects, POPE constructs triplets of the form:

$$\langle x, c, w_{(gt)} \rangle, \tag{13}$$

where $x$ represents the queried image, $c$ is the query prompt template, and $w_{(gt)}$ is the ground-truth answer to the query. The triplets generated by POPE include those with a "yes" response based on ground-truth objects and "no" responses obtained by sampling from negative objects. There are three strategies for negative sampling:

- **Random Sampling**: Randomly samples objects that do not exist in the image.
- **Popular Sampling**: Selects the top-$k$ most frequent objects in the image dataset that are absent from the current image.
- **Adversarial Sampling**: Ranks all objects based on their co-occurrence frequencies with the ground-truth objects and selects the top-$k$ frequent ones that do not exist in the image.

POPE employs the following evaluation metrics to measure performance:

$$\text{Accuracy} = \frac{\text{TP} + \text{TN}}{\text{TP} + \text{TN} + \text{FP} + \text{FN}}, \tag{14}$$

$$\text{Precision} = \frac{\text{TP}}{\text{TP} + \text{FP}}, \tag{15}$$

$$\text{Recall} = \frac{\text{TP}}{\text{TP} + \text{FN}}, \tag{16}$$

$$\text{F1 Score} = 2 \times \frac{\text{Precision} \times \text{Recall}}{\text{Precision} + \text{Recall}}. \tag{17}$$

In the above equations:

- **TP (True Positives)**: The number of correctly identified objects that are present in the image.
- **TN (True Negatives)**: The number of correctly identified objects that are absent from the image.
- **FP (False Positives)**: The number of objects incorrectly identified as present in the image.
- **FN (False Negatives)**: The number of objects that are present in the image but were not identified by the model.

These metrics provide a comprehensive evaluation of the model's ability to accurately identify the presence or absence of objects, thereby quantifying the extent of hallucinations in LVMs.

## B.2 BEAF Evaluation

BEAF (BEfore and AFter) [53] extends the evaluation framework beyond the text-axis hallucination assessment of POPE by simultaneously considering both text- and vision-axes. Additionally, BEAF introduces change-aware metrics, enabling a more granular evaluation of object hallucinations. Similar to POPE, BEAF employs binary classification tasks using triplets; however, it accounts for more complex perceptual changes within the dataset.

**Dataset Definition** BEAF utilizes a dataset $G$ composed of tuples:

$$G = \{(X_o, X_m, C, W_o, W_m, E)\}_{i=1}^{|G|}, \tag{18}$$

where $X_o$ denotes the original image. $X_m$ represents the change-aware manipulate image. $C$ is the question. $W_o$ and $W_m$ are the corresponding answers for the original and manipulated images, respectively. $E \in \{\text{True}, \text{False}\}$ indicates whether the question pertains to an object that has been removed in the manipulated image.

**Filter Function** To facilitate the extraction of specific subsets from $G$ based on input conditions, BEAF defines a filter function:

$$\text{Filter}(b_o, b_m, b_r) = \{h \mid \text{IsCorrect}(W_o) = bo, \text{IsCorrect}(W_m) = b_m, E = b_r, h \in G\}, \tag{19}$$

where $h = (X_o, X_m, C, W_o, W_m, E)$. Here, $b_o$, $b_m$, and $b_r$ are boolean values $\{\text{True}, \text{False}\}$ that specify the desired correctness and relation flags for filtering.

**Evaluation Metrics** Based on the `Filter` function, BEAF defines the following fine-grained perceptual change metrics:

$$\text{TU} = \frac{|\text{Filter}(\text{True, True, True})|}{|\text{Filter}(\text{True} \vee \text{False, True} \vee \text{False, True})|} \times 100, \tag{20}$$

$$\text{IG} = \frac{|\text{Filter}(\text{False, False, True})|}{|\text{Filter}(\text{True} \vee \text{False, True} \vee \text{False, True})|} \times 100, \tag{21}$$

$$\text{SB}_p = \frac{|\text{Filter}(\text{True, False, True})|}{|\text{Filter}(\text{True} \vee \text{False, True} \vee \text{False, True})|} \times 100, \tag{22}$$

$$\text{SB}_n = \frac{|\text{Filter}(\text{False, True, True})|}{|\text{Filter}(\text{True} \vee \text{False, True} \vee \text{False, True})|} \times 100, \tag{23}$$

$$\text{ID} = \frac{|\text{Filter}(\text{True, False, False})| + |\text{Filter}(\text{False, True, False})|}{|\text{Filter}(\text{True} \vee \text{False, True} \vee \text{False, False})|} \times 100, \tag{24}$$

$$\text{F1}_{\textbf{TUID}} = \frac{2 \times \text{TU}}{1 + (100 - \text{ID})}, \tag{25}$$

where TU represents True Understanding, IG denotes Ignorance, SB refers to Stubbornness, and ID signifies Indecision. These metrics provide a more nuanced evaluation of the model's capacity to recognize and adapt to perceptual changes across textual and visual contexts, offering a comprehensive assessment of hallucinations in LVMs.

## C  More Details of Experiment Results

### C.1  Evaluation of Text-Axis and Vision-/Text-Axis Hallucinations

Table 8 presents the performance evaluation of Precision (Prec.) and Recall under the POPE and BEAF experimental settings. The results demonstrate that our method achieves effective improvements in both text-axis and vision-/text-axis hallucination evaluations. While a slight decrease in Recall is observed in some cases, the overall performance exhibits significant enhancement. Notably, the decline in Recall is minimal, whereas the improvement in Precision is more pronounced, further validating the effectiveness of our approach.

Table 8: Comparison of text-axis evaluation across three POPE evaluation settings: Random Sampling, Popular Sampling, and Adversarial Sampling on the MSCOCO validation set. Additionally, vision- and text-axis evaluations are conducted under the BEAF benchmark. The values highlighted in green represent the percentage improvements achieved by our proposed method, whereas the values in red indicate performance degradation.

| LVM | Vision Input | POPE-Popular | | POPE-Random | | POPE-Adversarial | | BEAF | |
| --- | --- | --- | --- | --- | --- | --- | --- | --- | --- |
| | | Prec.↑ | Recall↑ | Prec.↑ | Recall↑ | Prec.↑ | Recall↑ | Prec.↑ | Recall↑ |
| LLaVA-v1.5 | *Original* | 82.87 | 90.09 | 88.13 | 90.07 | 74.45 | 90.73 | 61.77 | 92.43 |
| | +*VAP* | $\mathbf{83.95}^{+1.08}$ | $\mathbf{90.67}^{+0.58}$ | $\mathbf{89.47}^{+1.34}$ | $\mathbf{90.67}^{+0.60}$ | $\mathbf{75.27}^{+0.82}$ | $\mathbf{92.04}^{+1.31}$ | $\mathbf{62.32}^{+0.55}$ | $92.13^{-0.30}$ |
| Instruct-BLIP | *Original* | 85.15 | 80.67 | 94.83 | 80.67 | 82.21 | 81.33 | 67.00 | 81.52 |
| | +*VAP* | $\mathbf{85.78}^{+0.63}$ | $\mathbf{81.67}^{+1.00}$ | $\mathbf{95.70}^{+0.87}$ | $\mathbf{81.67}^{+1.00}$ | $\mathbf{82.50}^{+0.29}$ | $\mathbf{82.42}^{+1.09}$ | $\mathbf{67.47}^{+0.47}$ | $\mathbf{81.83}^{+0.31}$ |
| Intern-VL2 | *Original* | 95.62 | 71.90 | 97.40 | 71.71 | 92.50 | 71.64 | 87.40 | 72.24 |
| | +*VAP* | $\mathbf{97.41}^{+1.59}$ | $\mathbf{74.13}^{+2.23}$ | $\mathbf{98.07}^{+0.67}$ | $\mathbf{73.58}^{+1.87}$ | $\mathbf{94.50}^{+2.00}$ | $\mathbf{73.66}^{+2.02}$ | $\mathbf{88.76}^{+1.36}$ | $\mathbf{72.35}^{+0.09}$ |
| Intern-VL2-MPO | *Original* | 93.70 | 80.39 | 95.39 | 80.95 | 90.55 | 81.08 | 82.46 | 82.67 |
| | +*VAP* | $\mathbf{94.11}^{+0.41}$ | $\mathbf{83.12}^{+2.73}$ | $\mathbf{96.48}^{+1.09}$ | $\mathbf{83.12}^{+2.17}$ | $\mathbf{91.62}^{+1.07}$ | $\mathbf{83.83}^{+2.75}$ | $\mathbf{83.52}^{+1.06}$ | $\mathbf{82.73}^{+0.06}$ |
| DeepSeek-VL2 | *Original* | 92.46 | 80.13 | 96.70 | 80.13 | 91.06 | 80.67 | 84.11 | 80.90 |
| | +*VAP* | $\mathbf{93.52}^{+1.06}$ | $\mathbf{80.80}^{+0.67}$ | $\mathbf{97.34}^{+0.64}$ | $\mathbf{80.81}^{+0.68}$ | $\mathbf{92.39}^{+1.33}$ | $\mathbf{80.93}^{+0.26}$ | $\mathbf{85.12}^{+1.01}$ | $\mathbf{81.21}^{+0.31}$ |
| Qwen-VL2 | *Original* | 91.15 | 84.47 | 96.28 | 84.47 | 87.21 | 84.87 | 78.62 | 83.81 |
| | +*VAP* | $\mathbf{92.34}^{+1.19}$ | $\mathbf{85.26}^{+0.79}$ | $\mathbf{97.39}^{+1.11}$ | $\mathbf{84.60}^{+0.13}$ | $\mathbf{88.87}^{+1.66}$ | $\mathbf{85.25}^{+0.38}$ | $\mathbf{80.03}^{+1.41}$ | $83.14^{-0.67}$ |
| LLaVA-OV | *Original* | 95.20 | 80.67 | 98.06 | 80.67 | 92.72 | 80.67 | 87.58 | 81.69 |
| | +*VAP* | $\mathbf{96.97}^{+1.77}$ | $\mathbf{80.81}^{+0.14}$ | $\mathbf{99.00}^{+0.94}$ | $80.56^{-0.11}$ | $\mathbf{93.54}^{+0.82}$ | $\mathbf{81.13}^{+0.46}$ | $\mathbf{88.17}^{+0.59}$ | $\mathbf{82.06}^{+0.37}$ |
| Ovis1.6-Gemma2 | *Original* | 95.45 | 79.72 | 97.87 | 79.65 | 91.19 | 80.16 | 86.17 | 80.95 |
| | +*VAP* | $\mathbf{96.74}^{+0.29}$ | $79.70^{-0.02}$ | $\mathbf{98.44}^{+0.57}$ | $\mathbf{80.45}^{+0.80}$ | $\mathbf{91.69}^{+0.50}$ | $\mathbf{81.03}^{+0.87}$ | $\mathbf{86.92}^{+0.75}$ | $\mathbf{82.27}^{+1.32}$ |

## C.2  Parameter Sensitive Analysis

Table 9 presents the parameter sensitivity analysis of the adversarial strategies loss function, as the parameters used in our approach vary across different models due to their distinct characteristics. The results indicate that parameter choices significantly impact performance metrics, including Accuracy (Acc.), Precision (Prec.), Recall (Rec.), and F1-score (F1). Notably, the selection of $\sqrt{1/\sigma_1}$, $\sqrt{1/\sigma_2}$, and $\sqrt{1/\sigma_3}$ involves a trade-off process, where optimizing one metric may lead to compromises in others. Interestingly, certain parameters yield competitive performance even when set to zero, suggesting potential redundancy in specific configurations. This trade-off underscores the necessity of carefully balancing parameter choices to achieve optimal overall performance.

Table 9: Parameter analysis of the Intern-VL2 [7] under varying settings of $\sigma_1$, $\sigma_2$, and $\sigma_3$. The model parameters were fixed as $\sqrt{1/\sigma_1} = 1.0$, $\sqrt{1/\sigma_2} = 0.5$, and $\sqrt{1/\sigma_3} = 0.5$ without changing the values of $\sigma_1$, $\sigma_2$, and $\sigma_3$. Performance comparison under the POPE Random evaluation setting, which involves randomly sampling objects that do not exist in the image. We randomly selected 1000 images from the MS-COCO dataset for this evaluation.

| Value | $\sqrt{1/\sigma_1}$ | | | | $\sqrt{1/\sigma_2}$ | | | | $\sqrt{1/\sigma_3}$ | | | |
|---|---|---|---|---|---|---|---|---|---|---|---|---|
| | Acc. ↑ | Prec. ↑ | Rec. ↑ | F1 ↑ | Acc. ↑ | Prec. ↑ | Rec. ↑ | F1 ↑ | Acc. ↑ | Prec. ↑ | Rec. ↑ | F1 ↑ |
| 0.0 | 87.20 | 95.72 | 77.24 | 85.49 | 86.82 | 95.65 | 76.38 | 84.94 | 87.54 | 94.95 | 78.47 | 85.93 |
| 0.1 | 86.77 | 95.61 | 76.22 | 84.82 | 87.75 | 96.52 | 77.62 | 86.04 | 86.82 | 95.65 | 76.38 | 84.94 |
| 0.25 | 86.73 | 94.78 | 76.76 | 84.82 | 87.83 | 95.76 | 78.47 | 86.25 | 87.45 | 94.87 | 78.16 | 85.71 |
| 0.5 | 87.45 | 95.68 | 77.62 | 85.71 | 88.09 | 96.55 | 78.32 | 86.48 | 87.79 | 95.72 | 78.32 | 86.15 |
| 0.75 | 87.24 | 94.95 | 77.93 | 85.60 | 87.83 | 94.95 | 79.02 | 86.25 | 87.58 | 95.79 | 78.08 | 86.03 |
| 1.0 | 87.92 | 95.80 | 78.62 | 86.36 | 87.50 | 95.72 | 77.77 | 85.82 | 87.58 | 95.79 | 78.08 | 86.03 |

## C.3  Impact of visual adversarial perturbation and uncertainty

Figure 5 show how model performance varies with different perturbation strengths ($\epsilon$) and distortion levels ($T$). We observe that performance initially improves with moderate perturbations, peaking before declining as perturbations grow stronger. When $\epsilon \geq 16$ or when $T$ leads to full Gaussian noise, performance drops below the no-VAP baseline. This indicates that (1) VAP effectively mitigates hallucinations by reducing semantic similarity between responses to original and distorted views under both conditional ($c$) and unconditional ($\emptyset$) settings, and (2) excessive perturbation harms visual feature extraction, undermining the model's ability to quantify parametric knowledge bias and ultimately degrading performance.

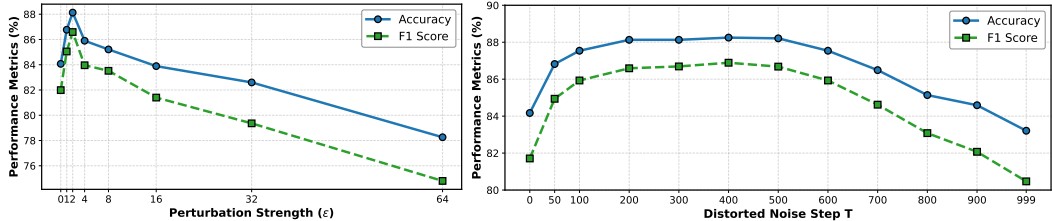

Figure 5: Performance of Intern-VL2 [7] under varying perturbation and distortion levels under POPE setting. The model is tested with varying perturbations applied to the original and distorted images.

## C.4  Ablation Study

Table 10 explore the effects of various combinations of loss functions ($\mathcal{L}_{s_1}$, $\mathcal{L}_{s_2}$, $\mathcal{L}_{s_3}$) on the performance of the Intern-VL2 model under the POPE evaluation setting. The results, as presented in Table 10, indicate that the simultaneous application of all three loss functions yields the highest accuracy and F1 score, achieving 84.81% and 82.79%, respectively. This suggests a synergistic effect when combining these losses, enhancing the model's ability to generalize effectively. Notably, the combination of $\mathcal{L}_{s_1}$ and $\mathcal{L}_{s_2}$ also shows a significant improvement over using any single loss function, highlighting the importance of multi-faceted optimization strategies.

Table 10: Impact of Different Loss Combinations on Model Performance: Ablation Study of Intern-VL2 Using the POPE Evaluation Setting.

| $\mathcal{L}_{s_1}$ | $\mathcal{L}_{s_2}$ | $\mathcal{L}_{s_3}$ | Acc.↑ | F1↑ |
|---|---|---|---|---|
| | | | 82.00 | 80.70 |
| ✓ | | | 83.07 | 81.55 |
| | ✓ | | 82.41 | 81.10 |
| | | ✓ | 82.36 | 81.04 |
| ✓ | ✓ | | 84.12 | 82.19 |
| ✓ | | ✓ | 84.05 | 82.08 |
| | ✓ | ✓ | 82.66 | 81.23 |
| ✓ | ✓ | ✓ | **84.81** | **82.79** |

# D  Generalization of VAP

The high computational cost of optimizing adversarial strategies poses a significant challenge [20, 44, 1, 19]. A practical approach to mitigate this challenge is to leverage smaller-scale models as proxies to generate visual perturbations. Table 11 demonstrates the strong generalization capability of VAP, where perturbations generated by smaller models effectively enhance the performance of larger counterparts. Specifically, applying perturbations from the Intern-VL2-1B model to Intern-VL2-8B results in a 1.78% improvement in F1 score, while substantially reducing inference costs—requiring only $\frac{1}{8}$ of the A100 computation time per sample compared to Intern-VL2-8B. A similar pattern is observed in the Qwen-VL2 series, where proxy-generated noise also leads to consistent performance improvements in larger-scale models. Although the performance gains from proxy-based perturbations are slightly lower than those from target model-generated noise, they provide an effective balance between computational efficiency and performance enhancement. These findings underscore the potential of VAP in scaling hallucination suppression across models of different sizes, offering a scalable and resource-efficient solution for real-world applications.

Table 11: Generalization performance of VAP across different models. The table compares the results obtained from the original images (left value) and the perturbed images generated using source models under the VAP setting (right value). Experiments are conducted on Intern-VL2 and Qwen-VL2 models, with the best results highlighted in **bold**. The inference cost reduction, shown in the last row, is measured relative to using the original target models.

| Metric | Source: Intern-VL2-1B | | | Source: Qwen-VL2-2B | |
|---|---|---|---|---|---|
| | ⇒ Intern-VL2-1B | ⇒ Intern-VL2-4B | ⇒ Intern-VL2-8B | ⇒ Qwen-VL2-2B | ⇒ Qwen-VL2-7B |
| Accuracy | 81.69/**83.28** | 81.55/**82.56** | 82.00/**84.07** | 84.47/**85.42** | 86.27/**86.87** |
| Precision | 89.72/**92.13** | 85.65/**87.21** | 87.40/**90.97** | 83.98/**84.85** | 87.21/**88.03** |
| Recall | 70.94/**72.34** | 75.05/**75.90** | 72.24/**75.50** | 84.04/**85.26** | 84.87/**85.33** |
| F1 Score | 79.23/**81.04** | 80.00/**81.16** | 80.70/**82.52** | 84.01/**85.05** | 86.02/**86.66** |
| Inference Cost Reduction | **1×** | **1/3×** | **1/8×** | **1×** | **1/5×** |

# E   Additional Illustration of Hallucination Evaluation

Figure 6 presents comprehensive hallucination evaluation examples from eight state-of-the-art LVMs, demonstrating the effectiveness of our proposed method across diverse model types. While different models exhibit varying response behaviors, our approach consistently mitigates hallucinations across all cases. Notably, in models such as Intern-VL2-MPO and Ovis1.6-Gemma2, our method not only corrects erroneous responses but also facilitates the generation of more factually accurate reasoning. Moreover, our observations reveal that certain models exhibit fixed template-like responses to queries, such as LLaVA-OV, which provides binary responses devoid of visual context. This characteristic underscores the challenges in improving performance for such models, as their outputs of this nature pose difficulties in adversarial optimization scenarios. These results substantiate the effectiveness of the introduced visual noise VAP in alleviating hallucinations during the inference process, helping LVMs to achieve more reliable and content-aware predictions by reducing their reliance on spurious correlations and enhancing their focus on visually grounded evidence.

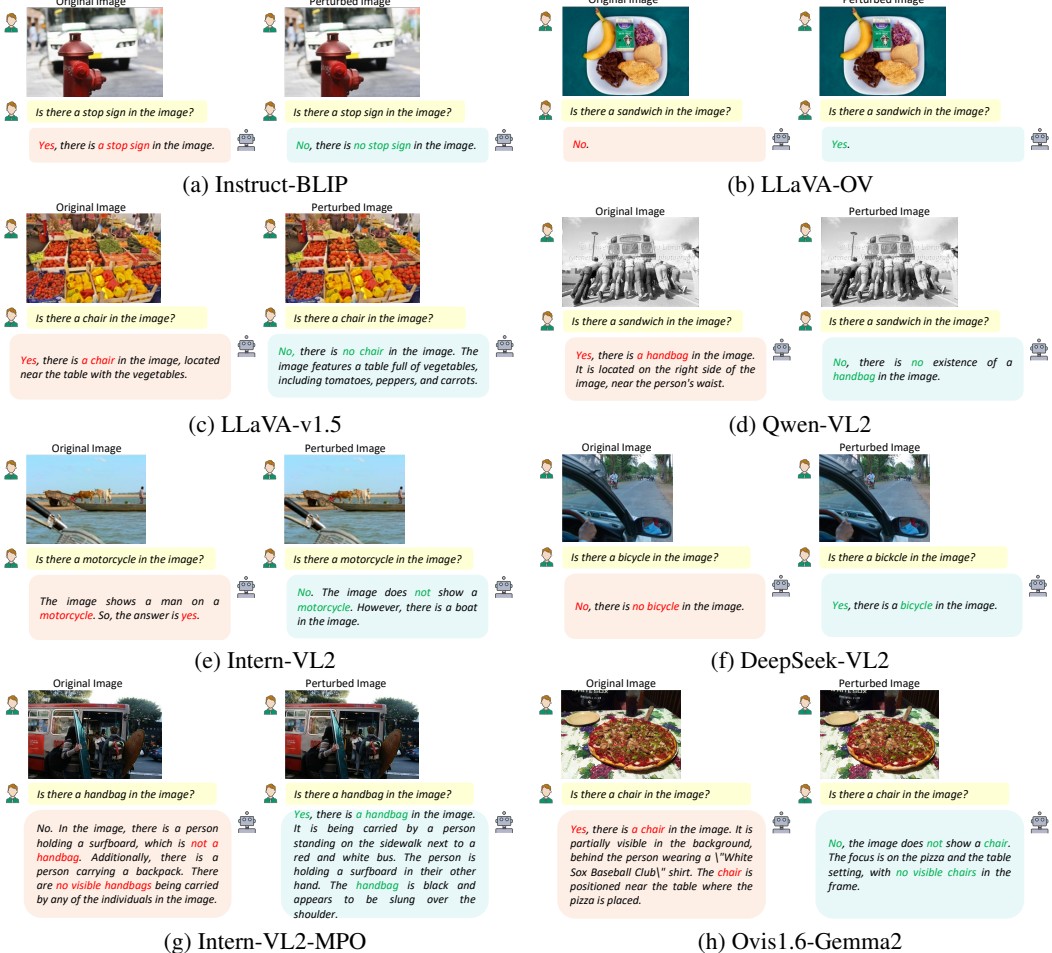

Figure 6: Illustrative examples from the POPE hallucination evaluation across eight large vision-language models: (a) Instruct-BLIP, (b) LLaVA-OV, (c) LLaVA-v1.5, (d) Qwen-VL2, (e) Intern-VL2, (f) DeepSeek-VL2, (g) Intern-VL2-MPO, and (h) Ovis1.6-Gemma2. The figure presents representative comparisons between original images and perturbed images enhanced with VAP, highlighting the differences in model responses.

# F   Orthogonality and Complementarity with Existing Methods

Unlike conventional model-centric approaches, our proposed method introduces a novel paradigm for hallucination mitigation by exploiting the very mechanisms responsible for hallucinations to suppress them. This strategy offers a fresh perspective on aligning parametric knowledge with visual evidence in large vision-language models (LVMs).

To verify the orthogonality and compatibility of VAP with existing methods, we integrate it with both OPERA [18], a recent state-of-the-art suppression approach, and VCD, another competitive baseline. As shown in Table 12, across four strong LVMs and three evaluation settings (POPE, BEAF, CHAIR), VAP consistently provides complementary gains. For example, on LLaVA-v1.5, *VAP + OPERA* reduces CHAIR$_S$ from 6.90 (*Regular*) to 6.10, while *VAP + VCD* achieves an even lower 5.80 with improved F1$_{TUID}$. Similar compounded benefits are observed on Intern-VL2, where TU rises from 64.12 (*Regular*) to 66.78 (*VAP + VCD*). Although margins vary across models (e.g., smaller gains on DeepSeek-VL2), the consistent trend demonstrates that VAP operates along an orthogonal axis and integrates effectively with both prior works.

In summary, VAP is methodologically orthogonal to existing strategies, intervening at the visual input level rather than architectural or loss modifications, and delivers non-redundant improvements when combined with strong baselines such as OPERA and VCD. This establishes a practical path for compounded effectiveness in future hallucination suppression systems.

Table 12: Comparison of hallucination suppression performance across four LVMs (LLaVA-v1.5, Qwen-VL2, Intern-VL2, DeepSeek-VL2) under three evaluation settings: POPE, BEAF, and CHAIR.

| LVM | Method | POPE | | BEAF | | CHAIR | |
|---|---|---|---|---|---|---|---|
| | | Acc. ↑ | F1 ↑ | TU ↑ | F1$_{TUID}$ ↑ | CHAIR$_I$ ↓ | CHAIR$_S$ ↓ |
| LLaVA-v1.5 | *Regular* | 79.80 | 81.79 | 34.25 | 50.31 | 4.01 | 6.90 |
| | *VCD* | 81.26 | 83.12 | 34.62 | 50.85 | 3.91 | 6.20 |
| | *OPERA* | 80.32 | 81.92 | 34.51 | 50.48 | 3.95 | 6.70 |
| | *VAP* | 80.97 | 82.82 | 34.83 | 50.97 | 3.86 | 6.10 |
| | *VAP + VCD* | **82.35** | **83.54** | **35.21** | **51.40** | **3.62** | **5.80** |
| | *VAP + OPERA* | 81.45 | 83.40 | 35.22 | 51.43 | 3.72 | 6.10 |
| Qwen-VL2 | *Regular* | 86.27 | 86.02 | 54.78 | 69.78 | 3.45 | 6.20 |
| | *VCD* | 87.60 | 87.25 | 56.12 | 71.05 | 3.18 | 6.00 |
| | *OPERA* | 86.68 | 86.42 | 55.34 | 70.18 | 3.36 | 6.00 |
| | *VAP* | 87.30 | 87.02 | 56.18 | 70.79 | 3.23 | 5.70 |
| | *VAP + VCD* | **87.55** | **87.18** | **56.40** | **70.91** | **3.11** | **5.50** |
| | *VAP + OPERA* | 87.40 | 87.12 | 56.32 | 70.89 | 3.21 | 5.60 |
| Intern-VL2 | *Regular* | 82.00 | 80.70 | 64.12 | 76.17 | 5.14 | 9.50 |
| | *VCD* | 84.32 | 82.30 | 65.88 | 77.43 | 4.72 | 9.00 |
| | *OPERA* | 83.12 | 81.54 | 64.93 | 76.75 | 4.94 | 9.20 |
| | *VAP* | 84.81 | 82.79 | 66.15 | 77.63 | 4.65 | 8.90 |
| | *VAP + VCD* | **85.60** | **83.41** | **66.78** | **78.34** | **4.41** | **8.50** |
| | *VAP + OPERA* | 85.09 | 83.00 | 66.35 | 77.78 | 4.60 | 8.80 |
| DeepSeek-VL2 | *Regular* | 86.47 | 85.55 | 67.04 | 79.27 | 1.84 | 4.50 |
| | *VCD* | 86.65 | 85.72 | 67.25 | 79.43 | 1.80 | 4.40 |
| | *OPERA* | 86.73 | 85.84 | 67.47 | 79.57 | 1.77 | 4.40 |
| | *VAP* | 87.13 | 86.28 | 68.11 | 80.03 | 1.66 | 4.30 |
| | *VAP + VCD* | 87.18 | 86.32 | 68.18 | 80.08 | 1.65 | 4.20 |
| | *VAP + OPERA* | **87.20** | **86.35** | **68.22** | **80.11** | **1.64** | **4.20** |

# G   Dynamics under Visual Uncertainty

To further understand how hallucinations evolve under degraded vision, we progressively injected Gaussian noise ($T$) into the inputs of Intern-VL2 and tracked two key indicators: S2 (prompt-driven hallucination) and S3 (prior-driven hallucination).

As shown in Table 13, without VAP the S2/S3 values remain relatively static, confirming that baseline models lack an uncertainty-aware mechanism to self-correct hallucinations. By contrast, with VAP the largest gains appear at moderate noise levels ($T \approx 200$), where input degradation is sufficient to trigger hallucinations but still informative for grounding.

At low noise levels ($T \leq 100$), the model is already well grounded and improvements are minor, while at high noise levels ($T \geq 500$) the input becomes too corrupted, leading to diminishing returns. These results demonstrate that VAP leverages uncertainty to suppress prompt- and prior-driven hallucinations, and is effective in realistic scenarios where vision is degraded but not lost.

Table 13: Dynamics of hallucination suppression under varying levels of visual uncertainty (Intern-VL2). VAP achieves maximal suppression at moderate noise ($T \approx 200$), confirming its ability to exploit uncertainty for robust grounding.

| Noise $T$ | S2 | | | S3 | | |
|---|---|---|---|---|---|---|
| | w/o ↓ | w/ ↓ | $\Delta$S2 ↑ | w/o ↓ | w/ ↓ | $\Delta$S3 ↑ |
| 0 | 0.75 | N/A | N/A | 0.79 | N/A | N/A |
| 100 | 0.73 | 0.64 | 0.09 | 0.76 | 0.69 | 0.07 |
| 200 | 0.70 | 0.60 | 0.10 | 0.74 | 0.66 | 0.08 |
| 300 | 0.68 | 0.59 | 0.09 | 0.72 | 0.65 | 0.07 |
| 500 | 0.65 | 0.57 | 0.08 | 0.70 | 0.64 | 0.06 |
| 700 | 0.61 | 0.53 | 0.08 | 0.65 | 0.60 | 0.05 |
| 999 | 0.45 | 0.40 | 0.05 | 0.50 | 0.47 | 0.03 |

## H Experimental Evaluation of Perturbation Perceptibility

To confirm that VAP introduces minimal visual distortion, we evaluate the perceptual similarity between original and perturbed images on 500 BEAF image–instruction pairs. Specifically, we measure LPIPS and SSIM, two widely used perceptual similarity metrics.

We consider four representative LVMs (LLaVA-v1.5, Qwen-VL2, Intern-VL2, and DeepSeek-VL2). For each image, we compute perceptual distances between the original image and: (a) its VAP-perturbed version, and (b) a Gaussian-noised version of the same magnitude.

The results reveal three key observations. First, VAP perturbations are visually negligible: all models achieve LPIPS $< 0.05$ and SSIM $> 0.95$, which aligns with standard perceptual quality thresholds. Second, VAP consistently yields lower LPIPS and higher SSIM than Gaussian noise, demonstrating superior perceptual fidelity. Finally, this confirms that VAP introduces only minimal distortion, thereby preserving visual utility and maintaining trust for real-world deployment.

Table 14: Perceptual similarity between original and perturbed images, measured by LPIPS (↓) and SSIM (↑). VAP perturbations remain visually negligible and consistently outperform Gaussian noise.

| Model | VAP Perturbation | | Gaussian Noise | |
|---|---|---|---|---|
| | LPIPS ↓ | SSIM ↑ | LPIPS ↓ | SSIM ↑ |
| LLaVA-v1.5 | 0.037 | 0.965 | 0.081 | 0.902 |
| Qwen-VL2 | 0.041 | 0.962 | 0.086 | 0.897 |
| Intern-VL2 | 0.039 | 0.967 | 0.079 | 0.906 |
| DeepSeek-VL2 | 0.035 | 0.969 | 0.077 | 0.911 |

## I Algorithm Details of VAP

Algorithm 1 outlines the procedure of our visual adversarial perturbation (VAP) method. VAP mitigates object hallucinations in LVMs by optimizing input perturbations that align model predictions more closely with visual evidence while reducing parametric bias. To handle the autoregressive nature of LVMs, we adopt a zeroth-order optimization strategy: sampling $N$ perturbations and approximating the gradient of the adversarial loss without accessing internal model parameters. The final perturbation is projected onto a bounded constraint $\mathbb{B}(\epsilon)$ before being applied, yielding perturbed inputs that effectively suppress hallucinations while preserving model usability.

**Algorithm 1** *Visual Adversarial Perturbation (VAP)*

---

**Adversarial Knowledge:** Image $x$, Query $c$, LVM $f_\theta$, Null text $\emptyset$, CLIP Text encoder $g_\psi$.
**Adversarial Setting:** Noise magnitude $\epsilon$, Distorted timestep $T$, Noise scheduling $\mu$, step size $\alpha$.
**Zero-Gradient Setting:** Number of queries $N$, Sampling variance $\beta$, Sampling noise $\gamma$.

1: Generate a distorted image:
$$\bar{x} \sim \mathcal{N}(\sqrt{\mu_T} x, (1 - \mu_T)\mathbf{I}). \tag{26}$$

2: Compute initial responses:
$$r_1^{(0)} = f_\theta(x, c), \quad r_2^{(0)} = f_\theta(x, \emptyset), \quad r_3 = f_\theta(\bar{x}, \emptyset). \tag{27}$$

3: Compute initial adversarial loss:
$$\mathcal{L}_{s_1}^{(0)} = \max g_\psi(r_1^{(0)})^\top g_\psi(r_2^{(0)}), \tag{28}$$
$$\mathcal{L}_{s_2}^{(0)} = \min g_\psi(r_1^{(0)})^\top g_\psi(r_3), \tag{29}$$
$$\mathcal{L}_{s_3}^{(0)} = \min g_\psi(r_2^{(0)})^\top g_\psi(r_3). \tag{30}$$

4: Compute overall initial loss:
$$\mathcal{L}_S^{(0)} = \frac{\mathcal{L}_{s_1}^{(0)}}{\sigma_1^2} + \frac{\mathcal{L}_{s_2}^{(0)}}{\sigma_2^2} + \frac{\mathcal{L}_{s_3}^{(0)}}{\sigma_3^2}. \tag{31}$$

5: **for** each zero-gradient optimization step $n \in \{1, \dots, N\}$ **do**
6:     Sample perturbation:
$$\gamma_n \sim P(\gamma), \text{ s.t. } \mathbb{E}[\gamma^\top \gamma] = I. \tag{32}$$

7:     Compute perturbed responses:
$$r_1^{(n)} = f_\theta(x + \beta \cdot \gamma_n, c), \tag{33}$$
$$r_2^{(n)} = f_\theta(x + \beta \cdot \gamma_n, \emptyset). \tag{34}$$

8:     Compute adversarial losses:
$$\mathcal{L}_{s_1}^{(n)} = \max g_\psi(r_1^{(n)})^\top g_\psi(r_2^{(n)}), \tag{35}$$
$$\mathcal{L}_{s_2}^{(n)} = \min g_\psi(r_1^{(n)})^\top g_\psi(r_3), \tag{36}$$
$$\mathcal{L}_{s_3}^{(n)} = \min g_\psi(r_2^{(n)})^\top g_\psi(r_3). \tag{37}$$

9:     Compute overall adversarial loss:
$$\mathcal{L}_S^{(n)} = \frac{\mathcal{L}_{s_1}^{(n)}}{\sigma_1^2} + \frac{\mathcal{L}_{s_2}^{(n)}}{\sigma_2^2} + \frac{\mathcal{L}_{s_3}^{(n)}}{\sigma_3^2}. \tag{38}$$

10: **end for**
11: Estimate perturbation direction via zeroth-order optimization:
$$\delta = \frac{1}{N \cdot \beta} \sum_{n=1}^{N} \{\mathcal{L}_S^{(n)} - \mathcal{L}_S^{(0)}\}. \tag{39}$$

12: Project perturbation onto $\delta \leftarrow \text{Proj}_{\mathbb{B}_\epsilon(x)}(\delta)$.
13: **Return response under VAP:**
$$w_{(VAP)} = f_\theta(\hat{x}, c) = f_\theta(x + \alpha \cdot \delta, c). \tag{40}$$

---

# J   Discussion

## J.1   Validation of Factual Comprehension

Our primary goal is to demonstrate that VAP does not impair the ability of models to comprehend factual content in images [14, 13, 55, 40]. Below, we present quantitative evaluations to substantiate this claim.

In Table 15, we provide evidence that VAP sustains and enhances model performance in factual object recognition and open-ended factual understanding tasks:

(1) Non-Hallucination Task Evaluation (MME [14]):

We evaluated four LVMs using the MME benchmark, which includes tasks such as existence detection, code reasoning, numerical calculations, and scene understanding. The results show that VAP maintains, and sometimes improves, accuracy in these factual and reasoning tasks. This confirms that VAP does not degrade performance on genuine questions.

(2) Multi-Dimensional Hallucination Grounding (AMBER [45]):

To assess generalization, we used the AMBER benchmark, which covers hallucinations in existence, attributes, and generative tasks. Our findings indicate that VAP enhances multi-dimensional visual grounding, further supporting its effectiveness without compromising factual understanding.

These evaluations collectively demonstrate that VAP enhances robustness while preserving the model's core perceptual and reasoning capabilities.

Table 15: Evaluation of VAP on MME and AMBER Benchmarks: Results show that VAP significantly enhances the models' abilities to accurately perceive, reason accurately, and ground visual content, confirming its effectiveness in reducing hallucinations while maintaining factual accuracy.

| LVM | Vision Input | MME (Perception and Reasoning) | | | | MME Total↑ | AMBER (Hallucination Analysis) | | |
|-----|-----|-----|-----|-----|-----|-----|-----|-----|-----|
| | | Exist.↑ | Code↑ | Cal↑ | Scene↑ | Score↑ | Cover↑ | Hal-Rate↓ | Cog↓ |
| LLaVA-v1.5 | *Original* | 93 | 50 | 40 | 83 | 982 | 51.7 | 35.4 | 4.2 |
| | *+VAP* | **95** | **55** | **43** | **86** | **1010** | **54.6** | **29.9** | **3.6** |
| Qwen-VL2 | *Original* | 95 | 78 | 73 | 81 | 1127 | 71.7 | 57.3 | 5.7 |
| | *+VAP* | **98** | **80** | **75** | **84** | **1169** | **72.8** | **54.1** | **4.9** |
| Intern-VL2 | *Original* | 90 | 75 | 60 | 83 | 1114 | 73.7 | 68.8 | 8.4 |
| | *+VAP* | **93** | **80** | **63** | **87** | **1146** | **75.2** | **65.8** | **7.5** |
| DeepSeek-VL2 | *Original* | 95 | 40 | 45 | 78 | 1024 | 48.2 | 9.5 | 0.4 |
| | *+VAP* | **98** | **45** | **48** | **81** | **1061** | **49.1** | **9.0** | **0.3** |

## J.2   Understanding the Effectiveness of VAP

The consistent performance improvements across different LVMs and evaluation frameworks raise an important question: why does VAP effectively mitigate hallucinations? Our analysis reveals key mechanisms underlying VAP's effectiveness:

**Balancing Visual and Language Signals**   The success of VAP can be primarily attributed to its ability to rebalance the interaction between visual and language processing in LVMs. This is evidenced by both the significant reduction in affirmative responses and performance improvements in vision-/text-axis hallucination assessments (Table 2). The BEAF evaluation framework particularly demonstrates how VAP effectively interrupts the model's default reliance on parametric knowledge. The carefully calibrated perturbations strengthen visual signals during the inference process, compelling the model to ground its responses more firmly in visual evidence rather than language priors.

**Adaptive Adversarial Noise Generation**   The effectiveness of VAP is further enhanced by its adaptive noise generation mechanism. Unlike traditional adversarial perturbations that aim to maximally disrupt model predictions, VAP generates "beneficial noise" through zero-gradient optimization that aligns response with grounding vision input and mitigates parametric knowledge bias. This selective enhancement is validated across multiple evaluation dimensions: (1) Closed VQA format

evaluations through both text-axis (POPE) and vision-/text-axis (BEAF) settings, and (2) Open-ended task evaluation through image caption generation (CHAIR). The consistent improvements across these diverse evaluation settings demonstrate VAP's ability to enhance visual understanding while maintaining task performance.

**Architecture-Agnostic Enhancement** Our experiments across different model architectures reveal that VAP's effectiveness is not tied to specific architectural choices. This architecture-agnostic nature can be explained by VAP's operation at the input level: it modifies the visual input distribution to better align with the model's learned visual-semantic mappings, regardless of the specific implementation details. This explanation is supported by the consistent performance improvements observed across models with varying architectures, ranging from pure transformer-based models to hybrid architectures across all three evaluation frameworks (POPE, BEAF, and CHAIR).

The combination of these mechanisms creates a powerful technique for hallucination mitigation:

- The rebalancing of visual-language interaction enhances visual perception while reducing spurious correlations stemming from biased language priors.
- The adaptive adversarial visual noise generation employs strategic optimization to influence LVM decision processes, ensuring that perturbations enhance rather than compromise visual understanding.
- VAP operates in a completely black-box manner requiring no access or modification to the LVM, establishing it as a broadly applicable solution across different model architectures.

