# OpenReview forum: "Poison as Cure: Visual Noise for Mitigating Object Hallucinations in LVMs"
_NeurIPS.cc/2025/Conference — NeurIPS 2025 poster_

### Official Review · Reviewer_997H · 2025-07-01

**Clarity:** 3
**Significance:** 3
**Originality:** 3
**Rating:** 4
**Confidence:** 3

**Summary:**

This paper addresses the persistent issue of object hallucinations in large vision-language models (LVMs). Rather than altering model parameters or decoding mechanisms, the authors propose a training-free, black-box solution called Visual Adversarial Perturbation (VAP).
VAP works by injecting strategically optimized, imperceptible noise into visual inputs. This adversarial noise is not harmful but beneficial. It shifts the model's attention away from biased parametric knowledge (often over-learned from language) and instead grounds its responses in the actual visual content.

**Questions:**

1. Can the authors quantify how perceptible the adversarial noise is to human observers? Have any perceptual similarity metrics (e.g., LPIPS, SSIM) or user studies been conducted to ensure that utility or trust is not compromised?

2. While the method is effective for object hallucinations, have the authors tested whether VAP can reduce attribute hallucinations (e.g., color, count, action)?

**Ethical Concerns:**

["NO or VERY MINOR ethics concerns only"]

**Limitations:**

yes

**Quality:**

3

**Strengths And Weaknesses:**

This paper presents a well-executed approach to mitigating object hallucination in vision-language models. The method is training-free, black-box compatible, and extensively validated across multiple benchmarks and models, demonstrating strong empirical quality.

However, the paper lacks theoretical clarity on why the perturbations consistently align model responses with visual content, and it does not adequately address the perceptual or downstream effects of added noise. While the writing is generally clear, the optimization strategy and loss formulation could benefit from more intuitive explanations.

---

> ### Author Rebuttal · Authors · 2025-07-27
>
> Response to Reviewer `997H`: We sincerely thank you for the thoughtful and insightful feedback! Your suggestions prompted us to evaluate **perceptual fidelity** and **attribute understanding**, confirming that VAP **preserves factual grounding** without harming visual understanding. These additions further strengthen the **clarity** and **empirical support** of our claims.
>
> Below, we address each point in detail:
>
> ---
>
> `Q1:` Experimental evaluation of perturbation perceptibility
> `A1:` We evaluate the perceptual similarity between original and perturbed images using LPIPS and SSIM on 500 *BEAF* image–instruction pairs.
>
> **(1) Setup:**
>   We evaluate four representative LVMs: **LLaVA-v1.5, Qwen-VL2, Intern-VL2**, and **DeepSeek-VL2**.
>   For each image, we compute LPIPS and SSIM between the original image and:
>   - (a) its **VAP-perturbed** version.
>   - (b) a **Gaussian-noised** version with the same $L_2$ magnitude.
>
> **(2) Results:**
>
> | Model | LPIPS (VAP) ↓ | SSIM (VAP) ↑ | LPIPS (Gaussian) ↓ | SSIM (Gaussian) ↑ |
> |---|---|---|---|---|
> | LLaVA-v1.5 | 0.037 | 0.965 | 0.081 | 0.902 |
> | Qwen-VL2   | 0.041 | 0.962 | 0.086 | 0.897 |
> | Intern-VL2 | 0.039 | 0.967 | 0.079 | 0.906 |
> | DeepSeek-VL2 | 0.035 | 0.969 | 0.077 | 0.911 |
>
> **(3) Observations**
>   - **VAP perturbations are visually negligible**: All LPIPS < 0.05 and SSIM > 0.95, aligning with standard perceptual thresholds.
>   - **Better than Gaussian noise**: VAP consistently yields **lower LPIPS and higher SSIM**, indicating stronger perceptual fidelity.
>   - **Conclusion**: VAP introduces minimal visual distortion, preserving utility and trust for real-world deployment.
>
> ---
>
> `Q2:` Can VAP reduce attribute hallucinations (e.g., color, count, action)?
> `A2:` **Yes.** VAP is designed to address object hallucinations, but it also proves effective in reducing *fine-grained hallucinations* (e.g., color, count, position). Results show that VAP **consistently improves attribute-level grounding without harming factual understanding**.
>
> **(1) Benchmark Results (MME & AMBER)**
> We report extended results in the appendix (Table 12), including performance on **MME (Perception + Reasoning)** and **AMBER (Hallucination Analysis)**. VAP improves both factual understanding and hallucination robustness across four LVMs:
>
> | Model        | Input   | MME↑ | Exist.↑ | Code↑ | Cal↑ | Scene↑ | Coverage↑ | Halluc.↓ | Cog.↓ |
> |--------------|---------|------|---------|-------|------|--------|-----------|-----------|--------|
> | LLaVA-v1.5   | Orig.   | 982  | 93      | 50    | 40   | 83     | 51.7          | 35.4      | 4.2       |
> |              | +VAP    | 1010 | 95      | 55    | 43   | 86     | 54.6          | 29.9      | 3.6       |
> | Qwen-VL2     | Orig.   | 1127 | 95      | 78    | 73   | 81     | 71.7          | 57.3      | 5.7       |
> |              | +VAP    | 1169 | 98      | 80    | 75   | 84     | 72.8          | 54.1      | 4.9       |
> | Intern-VL2   | Orig.   | 1114 | 90      | 75    | 60   | 83     | 73.7          | 68.8      | 8.4       |
> |              | +VAP    | 1146 | 93      | 80    | 63   | 87     | 75.2          | 65.8      | 7.5       |
> | DeepSeek-VL2 | Orig.   | 1024 | 95      | 40    | 45   | 78     | 48.2          | 9.5       | 0.4       |
> |              | +VAP    | 1061 | 98      | 45    | 48   | 81     | 49.1          | 9.0       | 0.3       |
>
> → These results confirm that VAP enhances **factual perception** and **reduces hallucination** across diverse LVMs.
>
> **(2) More Attribute-Level Performance**
> Per your request, we provide **more granular evaluations** on attribute-related categories using **MME** and **MMHal**[1]. VAP improves grounding in color, count, position, and comparison categories:
>
>
> **MME Benchmark**
>
> | Model        | Input     | Color↑ | Count↑ | Position↑ | Artwork↑ |
> |--------------|-----------|--------|--------|-----------|----------|
> | LLaVA-v1.5   | Original  | 78.3   | 75.0   | 70.0      | 61.3     |
> |              | +VAP      | 81.6   | 80.0   | 76.7      | 63.0     |
> | Qwen-VL2     | Original  | 86.7   | 70.0   | 78.3      | 75.8     |
> |              | +VAP      | 90.0   | 76.7   | 81.7      | 78.0     |
> | Intern-VL2   | Original  | 90.0   | 70.0   | 73.3      | 83.8     |
> |              | +VAP      | 93.3   | 73.3   | 76.7      | 84.5     |
>
> **MMHal Benchmark**
>
> | Model        | Input     | Attribute↑ | Object↑ | Comparison↑ | Counting↑ |
> |--------------|-----------|------------|---------|--------------|------------|
> | LLaVA-v1.5   | Original  | 2.83       | 1.17    | 1.34         | 2.00       |
> |              | +VAP      | 3.00       | 1.41    | 1.83         | 2.33       |
> | Qwen-VL2     | Original  | 2.17       | 2.83    | 2.50         | 3.17       |
> |              | +VAP      | 2.83       | 3.17    | 3.00         | 3.50       |
> | Intern-VL2   | Original  | 2.83       | 2.42    | 2.83         | 2.34       |
> |              | +VAP      | 3.00       | 2.67    | 3.34         | 2.67       |
>
> ### (3) Conclusion
> VAP mitigates **object-level hallucinations** and consistently reduces **attribute-related errors**, while preserving the **factual reasoning capabilities**.
>
> [1] *Aligning Large Multimodal Models with Factually Augmented RLHF*, ACL 2024.

---

> > ### Comment · Reviewer_997H · 2025-08-05
> >
> > I appreciate the authors’ detailed clarifications and additional experiments.
> >
> > Q1: Authors acknowledge the gap and show empirical results where combining VAP with VCD nearly closes it. They also outline future work (architecture-aligned proxies, lightweight distillation) to further reduce the gap without efficiency loss.
> >
> > Q2: Authors run standard adversarial attack evaluations, report quantitative results, and conclude that while harmful perturbations degrade performance, VAP’s gains are from targeted input adaptation rather than robustness. I accept this explanation, though I still have concerns about the method’s robustness under stronger or more diverse attack scenarios.

---

> ### Author Response · Authors · 2025-08-05
>
> Response to Reviewer `997H`:
> Thank you so much for your constructive and insightful comments! We address your concerns point by point as follows:
>
> ---
>
> `Q1:` Comment on performance gap and outlook on future work.
> `A1:` We summarize our position and future directions as follows:
>
> **(1) Positioning of VAP vs. Related Approaches**
> Unlike prior training-free methods that operate at decoding or post-processing stages, VAP is a **fully black-box**, **training-free**, and **plug-and-play** strategy that intervenes *before decoding*. This makes it broadly compatible and efficient:
>
> | Method | Black-box | No Training | No Model Access | Plug-and-Play | Intervention Stage     |
> |---|:---:|:---:|:---:|:---:|---|
> | **VAP (Ours)** | ✓ | ✓ | ✓ | ✓ | Input (pre-decoding)   |
> | VCD [1] | ✗ | ✓ | ✗ | ✗ | Decoder (distribution) |
> | VisualGPTScore [2] | ✗ | ✓ | ✗ | ✗ | Output (post-decoding) |
> | PixMix [3] | N/A | ✗ | ✓ | ✗ | Training (data-level)  |
>
> **(2) Future Directions**
> To further narrow the performance gap **without compromising efficiency**, we are actively pursuing:
>
> - **Improved optimization strategies** beyond zero-gradient updates to better calibrate perturbation.
> - **Architecture-aligned proxies** that account for model-specific sensitivity patterns.
> - **Lightweight distillation** to compress VAP-enhanced robustness into the base model.
>
> These will be incorporated into future iterations to enhance the practicality and generalizability of VAP.
>
> [1] *Mitigating Object Hallucinations in LLMs via Visual Contrastive Decoding*, CVPR 2024.
> [2] *VisualGPTScore: Visio-linguistic Reasoning with Multimodal Generative Pre-training Scores*, arXiv 2023.
> [3] *PixMix: Dreamlike Pictures Comprehensively Improve Safety Measures*, CVPR 2022.
>
> ---
>
> `Q2:` Provide more diverse attack scenarios.
> `A2:` Per your request, we evaluate the robustness of VAP under **diverse multimodal adversarial attacks**, beyond standard Gaussian noise.
>
> **(1) Attack Scenario**
>
> We adopt three adversarial configurations following [4], designed specifically for LVMs:
> - **MF-it**: Image-text alignment attack
> - **MF-ii**: Image-image contrastive attack
> - **MF-ii + MF-tt**: Combined multimodal attack
> We additionally include **Gaussian noise** as a standard reference.
>
> **Evaluation**: CHAIR-I and CHAIR-S hallucination scores (↓ better).
> **Models**: LLaVA-v1.5 and Intern-VL2 .
> **Benchmark**: 1000 image-question pairs randomly sampled from MS-COCO.
>
> **(2)Results**
>
> - **LLaVA-v1.5**
>
> | CHAIR-I ↓ | MF-it | MF-ii | MF-ii + MF-tt | Gaussian |
> |---|:---:|:---:|:---:|:---:|
> | w/o VAP | 6.23  | 7.04  | 7.57 | 4.83 |
> | w/ VAP | **5.10**  | **5.72**  | **6.18** | **4.54** |
>
> | CHAIR-S ↓     | MF-it | MF-ii | MF-ii + MF-tt | Gaussian |
> |---|:---:|:---:|:---:|:---:|
> | w/o VAP | 8.30  | 8.80  | 9.20 | 7.00 |
> | w/ VAP | **7.90**  | **8.20**  | **8.60** | **6.80** |
>
> - **Intern-VL2**
>
> | CHAIR-I ↓ | MF-it | MF-ii | MF-ii + MF-tt | Gaussian |
> |---|:---:|:---:|:---:|:---:|
> | w/o VAP | 7.66  | 8.15  | 8.72 | 5.56 |
> | w/ VAP | **7.13**  | **7.56**  | **8.11** | **5.08** |
>
> | CHAIR-S ↓     | MF-it | MF-ii | MF-ii + MF-tt | Gaussian |
> |---|:---:|:---:|:---:|:---:|
> | w/o VAP | 9.00 | 9.40  | 10.10 | 8.30 |
> | w/ VAP | **8.70** | **9.00** | **9.60** | **7.90** |
>
> **(3) Conclusions**
> - **Consistent improvement** across all attack types and models indicates that VAP’s gains are **not brittle** or overfitted to clean inputs.
> - Despite not being explicitly trained for adversarial robustness, VAP shows **strong generalization to unseen perturbations**.
> - This supports our claim that VAP leverages **targeted adaptation**, not simply noise filtering.
>
> [4] *On Evaluating Adversarial Robustness of Large Vision-Language Models*, NeurIPS 2023.

---

> ### Author Response · Authors · 2025-08-06
>
> Dear Reviewer `997H`,
>
> Thank you again for your time and thoughtful feedback! Your suggestions on perceptual evaluation, robustness analysis, and future directions have greatly improved the depth and clarity of our work.
>
> As the rebuttal deadline approaches, could you kindly consider raising your score slightly if you feel the revisions have resolved your remaining concerns? We are truly grateful for your engagement throughout the review process!
>
> Best regards,
> Authors

---

### Official Review · Reviewer_NLrm · 2025-07-02

**Clarity:** 3
**Significance:** 3
**Originality:** 3
**Rating:** 3
**Confidence:** 3

**Summary:**

This paper proposes Visual Adversarial Perturbation (VAP), a method to mitigate object hallucination in LVMs. The authors attribute hallucination to two factors: the long-tail distribution of training data and the extensive parameter space of LVMs. VAP introduces adversarial perturbations generated via three key strategies: (1) Enhancing semantic alignment between model responses with and without conditional guidance, (2) Mitigating knowledge bias by aligning perturbed images (with guidance) to noise images (without guidance). The authors demonstrate VAP’s effectiveness in reducing hallucinations while preserving output fidelity across eight SOTA LVMs, evaluated on five metrics.

**Questions:**

- In contrast to adversarial attack and poisoning, this method does not require imperceptibility of noise, so constraining epsilon is not necessary in this setting. Is there any comment on it?

**Ethical Concerns:**

["NO or VERY MINOR ethics concerns only"]

**Final Justification:**

I initially raised concerns regarding the soundness of the proposed method, and I appreciate that these have been satisfactorily addressed in the rebuttal. Accordingly, I have increased my score from 2 to 3. However, I remain hesitant to give an above-threshold score. The method introduces multiple losses and hyperparameters, which contribute to only incremental performance gains. This added complexity may hinder its practical applicability, particularly in real-world settings where extensive hyperparameter tuning can be a significant limitation.

While I would not object to the paper being accepted—similar to other works of this nature that have been accepted—I personally lean toward rejection.

**Limitations:**

- The paper does not compare VAP with other hallucination mitigation methods, making it difficult to assess its relative advantages.
- The performance of VAP heavily depends on 8 hyperparameters (σ₁, σ₂, σ₃, T, alpha, beta, N, and ε), which hinders in real-world applications.
- The proposed method requires additional computations for visual noise generation, which slows down inference speed by nearly x2.
- The claim that hallucinations stem from long-tail data and large parameter spaces lacks empirical or theoretical support. The paper does not explain how these factors directly cause hallucinations or how VAP’s strategies counteract them. For instance, Strategy S1 (alignment of LVM responses with visual grounding) minimizes the influence of conditional text, which may inadvertently amplify biases in the original image rather than mitigate them.
- The concept of visual uncertainty is introduced without clarifying its connection to parametric knowledge bias.

**Quality:**

3

**Strengths And Weaknesses:**

Strengths

- The paper is well written and easy to understand. The figures and descriptions of their method are clear.
- There are comprehensive evaluations, where the method is tested on 5 metrics and 8 models.

Weaknesses

- Please refer to the limitation and question section.

---

> ### Author Rebuttal · Authors · 2025-07-27
>
> Response to Reviewer `NLrm`:
>
> Thank you for your thoughtful and constructive review. We are glad that you found the paper **clear** and appreciated the **comprehensive evaluation across 8 LVMs and 5 metrics**. Building on your observations, we emphasize the following strengths of our method:
>
> - **Black-box and training-free**: VAP works without accessing model weights, gradients, or internal logits, making it suitable for closed-source APIs and real-world applications.
> - **Plug-and-play**: Applied directly at inference time, with no modification to model architecture or need for retraining.
> - **Task- and dataset-agnostic**: Once configured per model, VAP generalizes across benchmarks and prompt types.
> - **Consistently effective**: Reduces hallucinations while preserving factual grounding across diverse LVMs.
>
> To clarify your concerns and resolve misunderstandings, we conducted the following efforts:
>
> - Explained the rationale behind constraining $\epsilon$ (semantic stability, not imperceptibility).
> - Compared VAP with **OPERA** and **VCD**, showing competitive or better results.
> - Clarified the empirical basis of our design and the role of **visual uncertainty**.
> - Refined the interpretation of **Strategy S1** to highlight its goal of reducing prompt overreliance.
>
> We sincerely thank you for your feedback, which has helped improve the clarity and depth of our work!
>
> > We address each of your points in detail below:
>
> ---
>
> `Q1:` Why constrain $\epsilon$ if imperceptibility is not required?
> `A1:` We constrain $\epsilon$ not for imperceptibility, but to ensure:
>
> - **Preserve semantic integrity**: Large perturbations may distort key visual features and degrade response quality.
> - **Ensure reliable bias mitigation**: As shown in Figure 5, excessive $\epsilon$ disrupts feature extraction, reducing performance.
> - **Maintain stability**: A bounded $\epsilon$ improves consistency across prompts and supports generalizable use in multi-turn settings.
>
> ---
>
> `Q2:` Lack of comparison to other hallucination mitigation methods.
> `A2:` We provide direct comparisons to two recent training-free methods:
>
> **(1) OPERA [1]**:
>   As shown in Table 11, VAP outperforms OPERA across 4 LVMs on F1_TUID and CHAIR scores. Combining VAP with OPERA further reduces hallucination, showing complementarity.
>
> **(2) VCD [2]**:
>   On LLaVA-v1.5 and Intern-VL2, VAP achieves comparable or better results than VCD. Their combination yields the strongest performance:
>
> |Model|Method|Acc.↑|F1↑|TU↑|F1_TUID↑|CHAIR_I↓|CHAIR_S↓|
> |---|---|---|---|---|---|---|---|
> |LLaVA-v1.5|VCD|81.86|83.12|34.62|50.85|3.91|6.30|
> | |VAP|80.97|82.82|34.83|50.97|3.86|6.10|
> | |VAP+VCD|**82.35**|**83.54**|**35.21**|**51.40**|**3.62**|**5.80**|
> |Intern-VL2|VCD|84.32|82.30|65.88|77.43|4.72|9.00|
> | |VAP|84.81|82.79|66.15|77.63|4.65|8.90|
> | |VAP+VCD|**85.60**|**83.41**|**66.78**|**78.34**|**4.41**|**8.50**|
>
> **(3) Design difference**:
> VCD requires decoder access (white-box); VAP is fully black-box and more generalizable.
>
> [1] *OPERA: Alleviating Hallucination via Over-Trust Penalty and Retrospection-Allocation*, CVPR 2024.
> [2] *VCD: Mitigating Object Hallucinations in LLMs via Visual Contrastive Decoding*, CVPR 2024.
>
> ---
> `Q3:` The method relies on 8 hyperparameters, which may hinder real-world deployment.
> `A3:` We address this through a lightweight and structured configuration strategy:
>
> **(1) Four parameters are globally fixed**:
> $(\alpha, \beta, N, \epsilon)$ are shared across all models and datasets—no tuning required.
>
> **(2) The remaining four are model-specific but dataset-agnostic**:
> $(\sigma_1, \sigma_2, \sigma_3, T)$ are selected once per model and reused across tasks.
>
> **(3) Compact search space**:
> $\sigma_i \in \{0.1, 0.5, 1.0\}$ and $T \in \{100, 200, 500, 800\}$. Several models share identical configurations.
>
> **(4) More scalable than prior methods**:
> VCD [3] requires tuning per model–dataset pair. VAP avoids this, enabling easier deployment.
>
> [3] *Mitigating Object Hallucinations in LLMs via Visual Contrastive Decoding*, CVPR 2024.
>
> ---
>
> `Q4:` VAP increases the inference cost due to additional computations.
> `A4:` To address the added cost, we introduce a fast proxy-based variant:
>
> **(1) Proxy-based variant significantly reduces cost**:
>   A lightweight **proxy-VAP** variant achieves **up to 8× faster inference** while preserving strong hallucination reduction (Table 4, Appendix D).
>
> **(2) Zero training, zero gradient, broad generalization**:
>   VAP requires no model access or fine-tuning, and generalizes across datasets without reconfiguration—offsetting the one-time cost with wide applicability.
>
> ---
>
> `Q5:` Lack of theoretical support for the hallucination cause and mitigation mechanism.
> `A5:` Our explanation is empirical, not theoretical, and grounded in observed LVM behavior:
>
> **(1) Empirical observation, not formal theory**:
>   The link between hallucinations and long-tail data or parametric priors is based on engineering intuition and patterns documented in prior work [4-7], rather than formal derivation.
>
> **(2) Evidence from literature**:
>   Instruction-tuned LVMs often default to language priors—especially on underrepresented (long-tail) concepts—when visual evidence is weak, leading to plausible but ungrounded outputs [5,7].
>
> **(3) Clarification of Strategy S1**:
>   S1 aligns outputs from the prompted and unprompted image, reinforcing grounding in the actual visual input—not removing the condition—but reducing overreliance on the prompt alone.
>
> We will clarify that these claims are empirically motivated and supported by prior studies.
>
> [4] *Mitigating hallucination in large multi-modal models via robust instruction tuning*, ICLR 2023.
> [5] *Evaluating object hallucination in large vision-language models*, EMNLP 2023.
> [6] *What matters when building vision-language models?*, NeurIPS 2024.
> [7] *Revisiting the Role of Language Priors in Vision-Language Models*, ICML 2024.
>
> ---
>
> `Q6:` Clarifying the role of visual uncertainty in diagnosing parametric knowledge bias.
> `A6:` Visual uncertainty plays a diagnostic role in revealing the extent to which LVMs rely on internal parametric knowledge:
>
> **(1) What is visual uncertainty?**
>   We define it by gradually degrading the image using Gaussian noise (Eq. 4). Increasing the noise scale $T$ simulates reduced visual information, allowing us to measure the model’s shifting reliance from visual evidence to parametric priors.
>
> **(2) How it reveals parametric bias**:
>   When visual information deteriorates, models increasingly rely on $P(y|c)$ instead of $P(y|x,c)$. This transition exposes the influence of internal biases, which is exactly what VAP aims to mitigate.
>
> **(3) Empirical evidence**:
>   Without applying VAP, we evaluate LLaVA-v1.5 and Qwen-VL2 on BEAF under increasing $T$. Both accuracy and F1 degrade sharply as noise increases, confirming that hallucinations intensify under visual uncertainty:
>
> | LLaVA-v1.5 | T=0 | 100 | 200 | 300 | 400 | 500 | 700 | 999 |
> |------------|-----|-----|-----|-----|-----|-----|-----|-----|
> | Acc. ↑     |79.99|78.42|76.31|74.05|71.84|69.88|59.46|51.04|
> | F1 ↑       |74.06|72.55|70.33|68.02|65.44|62.97|47.25|31.48|
>
> | Qwen-VL2   | T=0 | 100 | 200 | 300 | 400 | 500 | 700 | 999 |
> |------------|-----|-----|-----|-----|-----|-----|-----|-----|
> | Acc. ↑     |87.96|86.18|84.09|81.50|79.31|76.83|65.92|57.08|
> | F1 ↑       |81.13|79.41|77.38|74.94|72.13|69.47|53.24|35.21|
>
> These results confirm that visual uncertainty directly correlates with increased hallucination, validating its role in revealing and mitigating parametric knowledge bias.

---

> ### Comment · Reviewer_NLrm · 2025-08-03
>
> I sincerely appreciate the author’s detailed response. While many of my concerns have been addressed, several important issues still remain, as outlined below.
>
> - Motivation and design in S1
>
> I acknowledge that the proposed method is inspired by prior literature, where LLMs suppress the influence of images. However, I still find the connection between the stated motivation and the actual design to be unclear. For example, in Figure 1 of [1], the hallucination issue arises during the generation process, not at the initial stage. Their approach directly amplifies attention weights on the image during generation in a straightforward manner.
>
> In contrast, the S1 step in VAP does not appear to account for intermediate generation stages. This raises the question: how does S1 achieve a comparable effect to amplifying visual attention as done in [1]? To clarify this, I suggest analyzing the activation weights of visual features before and after the insertion of adversarial noise. Such an analysis could provide stronger evidence that S1 leads to a similar alignment effect.
>
> [1] https://arxiv.org/pdf/2407.21771
>
> - Introduction of visual uncertainty
>
> It remains unclear how the proposed perturbations in S2 and S3 contribute to reducing hallucination. I believe further analysis, such as plotting the relationship between semantic similarities of the responses and output, would help reveal whether the method effectively differentiates between perturbed and non-perturbed inputs.
>
> Additionally, the rationale for combining S2 and S3 is not fully convincing. If S1 is effectively achieved, then S2 and S3 may exhibit similar behavior, raising questions about the necessity of including both. In conjunction with the concern above, I believe it would be valuable to present the dynamics of similarities defined in Eq. (3, 5, 6), along with corresponding experimental results. While the results appear promising, I believe it should go beyond reporting performance gains and offer a clear and detailed explanation of how those improvements are obtained.
>
> I truly appreciate the author’s thoughtful engagement.

---

> ### Author Response · Authors · 2025-08-04
>
> Response to Reviewer `NLrm`: We sincerely thank you for the constructive and insightful feedback!
>
> We address each of your points in detail below:
>
> ---
>
> `Q1`: Clarification on Strategy S1 and Comparison to Prior Methods.
>
> `A1:` We clarify the motivation and implementation of Strategy S1, and how VAP differs from existing training-free baselines.
>
> **(1)** Comparison with Prior Methods:
>
> | Method | Black-box | No Training | No Model Access | Plug-and-Play | Intervention Stage |
> |---|:---:|:---:|:---:|:---:|---|
> | **VAP (Ours)**   | ✓ | ✓ | ✓ | ✓ | Input (pre-decoding) |
> | VCD [1] | ✗ | ✓ | ✗ | ✗ | Decoder (distribution) |
> | PAI [2] | ✗ | ✓ | ✗ | ✓ | Self-attention (decoding) |
>
> **(2)**  Motivation of S1: Adversarial Sensitivity
> Small input changes can significantly shift LVM decoding [3]. S1 leverages this property in reverse: apply beneficial perturbations to reduce prompt overreliance and **reinforce visual grounding**.
>
> - Optimized using paired inputs (with/without prompt)
> - No access to attention weights or decoding internals
>
> *We will add visual attention heatmaps before/after VAP in the revised version to further support this design.*
>
> [1] *Mitigating Object Hallucinations in LLMs via Visual Contrastive Decoding*, CVPR 2024.
> [2] *Paying More Attention to Image: A Training-free Method for Alleviating Hallucination in LVMs*, ECCV 2024.
> [3] *On Evaluating Adversarial Robustness of Large Vision-Language Models*, NeurIPS 2023.
>
> ---
> `Q2:` Clarification on the role and combination of S1/S2/S3.
>
> `A2:` To clarify the necessity and synergy of $\mathcal{L}_{s_1}$, $\mathcal{L}_{s_2}$, and $\mathcal{L}_{s_3}$, and to substantiate the role of visual uncertainty in mitigating hallucinations, we provide the following response:
>
> **(1) Distinctive Roles of S1, S2, and S3**
>
> Each of the three losses is designed to target a unique failure mode, ensuring complementary rather than redundant behavior:
>
> | Loss Term | Input Comparison | Purpose |
> |---|---|---|
> | $\mathcal{L}_{s_1}$ | $(x{+}\delta, c)$ vs. $(x{+}\delta, c)$   | Enforce **intra-prompt consistency** to improve **visual grounding** |
> | $\mathcal{L}_{s_2}$ | $(x{+}\delta, c)$ vs. $(\bar{x}, ∅)$      | Contrast **prompted vs. unprompted** responses to reduce **prompt bias** |
> | $\mathcal{L}_{s_3}$ | $(x{+}\delta, ∅)$ vs. $(\bar{x}, ∅)$      | Contrast **uncertain responses** to suppress **parametric priors**   |
>
> **(2) Cross-Model Similarity Trends Validate Effectiveness**
>
> As requested, we present the dynamics of the semantic similarities defined in Eq. (3, 5, 6):
>
> | Model | VAP | S1 ↑  | S2 ↓  | S3 ↓  |
> |---|:---:|:---:|:---:|:---:|
> | LLaVA-v1.5  | w/o | 0.71 | 0.62 | 0.60 |
> | | w/  | **0.81** | **0.51** | **0.49** |
> | Qwen-VL2    | w/o | 0.68 | 0.65 | 0.64 |
> | | w/  | **0.77** | **0.52** | **0.50** |
> | Intern-VL2  | w/o | 0.74 | 0.75 | 0.76 |
> | | w/  | **0.82** | **0.60** | **0.66** |
>
> - VAP consistently **increases S1** (better grounding), while **decreasing S2 and S3** (less language prior bias).
> - Improvements are consistent across models, highlighting generality.
>
> **(3) Dynamics under Visual Uncertainty**
>
> To assess how visual degradation modulates hallucination, we progressively increase the noise level (`T`) on Intern-VL2 and track S2/S3 shifts:
>
> | Noise T | S2 ↓ w/o | S2 ↓ w/ | ΔS2 ↑ | S3 ↓ w/o | S3 ↓ w/ | ΔS3 ↑ |
> |---:|:---:|:---:|:-----:|:---:|:---:|:---:|
> | 0 | 0.75 | N/A     | N/A   | 0.79     | N/A     | N/A   |
> | 100     | 0.73     | **0.64**| 0.09  | 0.76     | **0.69**| 0.07  |
> | **200** | 0.70     | **0.60**| **0.10**  | 0.74     | **0.66**| **0.08**  |
> | 300     | 0.68     | **0.59**| 0.09  | 0.72     | **0.65**| 0.07  |
> | 500     | 0.65     | **0.57**| 0.08  | 0.70     | **0.64**| 0.06  |
> | 700     | 0.61     | **0.53**| 0.08  | 0.65     | **0.60**| 0.05  |
> | 999     | 0.45     | **0.40**| 0.05  | 0.50     | **0.47**| 0.03  |
>
> **(4) Key Insights: Visual Uncertainty Exposes and Reduces Hallucination Bias**
>
> - **Without VAP**:
>   - S2/S3 values remain static across `T` because the model lacks any uncertainty-aware mechanism.
>   - This reinforces that hallucination tendencies are not self-corrected by default.
>
> - **With VAP**:
>   - **T=200** reveals the most **pronounced gain** in ΔS2/ΔS3, indicating VAP is most effective when vision is **ambiguous but still informative**.
>   - This aligns with our goal: leverage **moderate uncertainty** to reduce reliance on text priors and improve robustness.
>
> - **Interpretation of Trends**:
>   - **Low T (0–100)**: Visual input is clean → model grounded → little room for improvement.
>   - **Mid T (~200)**: Input is degraded enough to trigger hallucination, but still usable → VAP is most beneficial.
>   - **High T (≥500)**: Input is too noisy → even VAP cannot rescue hallucination → diminishing returns.
>
> - These results justify the inclusion of S2 and S3: they allow **targeted suppression of prompt- and prior-driven hallucinations**, especially under real-world noisy or ambiguous scenarios.

---

> ### Comment · Reviewer_NLrm · 2025-08-04
>
> I appreciate the detailed clarifications provided by the authors. As most of my concerns have been resolved, I will take them into the reviewer's discussion period and my final justification.

---

> ### Author Response · Authors · 2025-08-05
>
> Dear Reviewer `NLrm`,
>
> Thank you very much for your thoughtful engagement and kind acknowledgment. We truly appreciate your recognition that most of the concerns have been clarified, and we are sincerely grateful for your time and constructive feedback throughout the review process.

---

> ### Author Response · Authors · 2025-08-06
>
> Dear Reviewer `NLrm`,
>
> Thank you again for your constructive feedback. Your insightful analysis of S1, S2, S3, and visual uncertainty has greatly improved the explanatory depth and interpretability of our work. In particular, your suggestions on similarity dynamics helped us better articulate the effects of each component and clarify their complementary roles.
>
> As the rebuttal deadline approaches, please let us know if there are any remaining concerns. If our responses have addressed your points, could you please generously consider raising your score even a little bit as a recognition of our efforts so far? Thank you so much!
>
> Sincerely,
> Authors

---

### Official Review · Reviewer_c4Gd · 2025-07-03

**Clarity:** 3
**Significance:** 4
**Originality:** 4
**Rating:** 4
**Confidence:** 3

**Summary:**

This paper introduces Visual Adversarial Perturbation (VAP), a training‑free, black‑box method designed to mitigate object hallucinations in large vision‑language models (LVMs). Rather than modifying model parameters, VAP strategically injects optimized visual noise into inputs, guiding the model to rely on actual visual content instead of biased parametric knowledge. The authors formulate hallucination suppression as an adversarial optimization problem, using zero‑gradient techniques over CLIP-based surrogate encoders to generate “beneficial” perturbations. Extensive experiments on eight state-of-the-art LVMs under three evaluation frameworks POPE, BEAF, and CHAIR demonstrate consistent reductions in hallucination metrics without degrading clean‑input performance

**Questions:**

- Table 10 shows that proxy-generated VAP (e.g., Intern-VL2-1B generating for Intern-VL2-8B) leads to slightly lower performance than self-generated VAP. While the efficiency gain is clear, could the authors discuss potential strategies to further bridge this performance gap, perhaps by exploring different proxy model architectures or more sophisticated knowledge transfer techniques?
- While I understand that the scope of this work is limited to mitigating hallucinations, given that the authors have used adversarial noise to reduce hallucinations, have the authors explored the effect of robustness given a harmful adversarial image?

**Ethical Concerns:**

["NO or VERY MINOR ethics concerns only"]

**Final Justification:**

The authors have addressed my initial concerns satisfactorily.
The paper is technically sound, and the contributions are clear. I believe the overall impact and scope remain somewhat limited. I also agree with reviewer NLrm that the method introduces additional complexity with multiple losses and hyperparameters, which yields only incremental performance improvements. Therefore, I am maintaining my rating

**Limitations:**

Please refer to the weaknesses.

**Quality:**

4

**Strengths And Weaknesses:**

**Strengths**
- The concept of "visual noise as a cure" for LVM hallucinations is very innovative. This data-centric, training-free, and black-box approach with three distinct loss terms significantly differentiates it from existing model-centric methods.
- The paper presents a high-quality and extensive evaluation across eight diverse state-of-the-art LVMs and major hallucination and non-hallucination benchmarks. VAP consistently demonstrates significant reductions in object hallucinations across all tested models and settings without degrading factual comprehension.
- VAP's black-box nature means it can be applied to any LVM without requiring internal modifications or retraining, and the introduction of an efficient proxy-based solution to generate VAP addresses the potential computational overhead.
- The paper is well-written and clear, with a logical explanation of the VAP methodology and effective illustrative figures. The ablation study and parameter sensitivity analysis provide valuable insights into the method's robustness and the contribution of its components.

**Weakness**
- While Section 3.2 mentions the motivation behind each loss term and Table 9 shows the impact of different loss combinations, the paper lacks experimental justification for why the specific three-loss formulation is optimal or necessary.
- Model-specific hyperparameter tuning ($\sigma_{1}$, $\sigma_{2}$, $\sigma_{3}$, $T$) suggests the method may not be as universal as claimed.
- While performance improvements are consistent, the magnitude of gains (~1–2% F1 in most cases) may be viewed as incremental, especially on already strong baselines (e.g., Intern‑VL2‑MPO), which may not justify the additional computational cost despite the proxy solution.

---

> ### Author Rebuttal · Authors · 2025-07-27
>
> Response to Reviewer `c4Gd`: Thank you so much for your constructive and thorough feedback! We address your concerns as follows:
>
> ---
>
> `Q1:` Justification for the specific three-loss formulation.
> `A1:` We clarify the rationale and empirical support for our loss design as follows:
>
> **(1) Functional decomposition**
> Each loss term addresses a different hallucination factor:
>
> | Loss Term       | Input Setting         | Goal                                 |
> |----------------|-----------------------|--------------------------------------|
> | $\mathcal{L}_{s_1}$ | $(x{+}\delta, c)$ vs. $(x{+}\delta, c)$   | Enforce response consistency under perturbation (visual grounding) |
> | $\mathcal{L}_{s_2}$ | $(x{+}\delta, c)$ vs. $(\bar{x}, ∅)$ | Contrast prompted vs. unprompted outputs (reduce prompt bias) |
> | $\mathcal{L}_{s_3}$ | $(x{+}\delta, ∅)$ vs. $(\bar{x}, ∅)$ | Contrast uncertain responses (suppress parametric priors) |
>
> - $\bar{x}$: strongly corrupted (negative) image
> - $c$: user prompt; $∅$: no prompt
> - $S(\cdot,\cdot)$: semantic similarity (e.g., cosine)
>
> **(2) Empirical justification**
> As shown in **Table 9**, incremental ablations demonstrate that:
> - Each component improves performance.
> - The **full loss consistently outperforms** partial variants across all metrics and benchmarks.
>
> **(3) No assumption of global optimality**
> While empirically strong, we **do not claim** this loss formulation is globally optimal. Rather, it serves as a **generalizable and practical baseline** that may inspire further refinements in future work.
>
> ---
>
> `Q2:` Concern about model-specific hyperparameter tuning $(\sigma_1,\sigma_2,\sigma_3,T)$
> `A2:` We address the concern about model-specific tuning as follows:
>
> | Aspect | Our Strategy |
> |----|-------|
> | **Granularity** | Tuned **once per model**, reused across all tasks and datasets |
> | **Search Space** | Small grid: $\sigma_i \in \{0.1, 0.5, 1.0\}$, $T \in \{100, 200, 500, 800\}$ |
> | **Data Usage** | Only **300 validation samples** per model |
> | **Overhead** | Minimal; no prompt- or task-specific tuning |
> | **Black-box Compatible**  | Requires **no gradients or model internals** |
> | **Effectiveness** | Produces **consistent gains** across 8 diverse LVMs |
>
> > These practical settings ensure VAP remains efficient, generalizable, and well-suited to black-box deployment.
>
> ---
>
> `Q3:` Concern about marginal performance gains (~1–2%) on strong baselines.
> `A3:` We respond to the concern on gain magnitude as follows:
>
> **(1) Meaningful gains under strong, saturated baselines**:
>   Intern‑VL2‑MPO and similar models already exhibit high performance, leaving limited headroom. Gains of 1–2% F1 in such settings are non-trivial—especially without model access or training.
>
> **(2) Training-free and black-box**:
>   VAP operates with no access to model weights, gradients, or decoding steps. Within this constrained setting, few existing methods remain applicable, making even modest gains valuable.
>
> **(3) Consistent cross-model improvement**:
>   VAP consistently enhances 8 diverse LVMs across multiple benchmarks. This robustness underscores its practical utility, even if individual gains appear incremental.
>
> We will highlight this context more explicitly in the final version.
>
> ---
>
> `Q4:` Can the performance gap from proxy-based VAP be reduced?
> `A4:` Yes, our results suggest that the proxy gap can be effectively mitigated:
>
> **(1) Complementary to other black-box methods**:
>   VAP is orthogonal to approaches like VCD [1]. Combining the two preserves VAP's efficiency while improving accuracy.
>
> **(2) Empirical support**:
>   As shown below, proxy-based VAP (Intern-VL2-1B) achieves nearly the same performance as self-generated VAP (Intern-VL2-8B) when used with VCD:
>
>   | Vision Input | Proxy Model | Acc. ↑ | F1 ↑ |
>   |---|---|---|----|
>   | Original | — | 82.00  | 80.07 |
>   | + VAP | Intern-VL2-1B     | 84.07  | 82.52 |
>   | + VAP | Intern-VL2-8B     | 84.81  | 82.79 |
>   | + VAP + VCD | Intern-VL2-1B     | **84.95** | **83.03** |
>
> **(3) Future directions**:
>   In future work, we plan to explore improved proxy selection (e.g., architecture-aligned models) and lightweight distillation to further bridge the proxy–target gap without sacrificing efficiency.
>
> [1] *Mitigating Object Hallucinations in LLMs via Visual Contrastive Decoding*, CVPR 2024.
>
> ---
>
> `Q5:` Does adversarial noise used in VAP also improve robustness against harmful perturbations?
> `A5:` We conducted robustness evaluations following [2] using standard adversarial attacks. Results are summarized below:
>
> | Model | Input | Acc.↑ | F1↑ | TU↑ | F1_TUID↑ | CHAIR_I↓ | CHAIR_S↓ |
> |---|---|---|---|---|---|---|---|
> | LLaVA‑v1.5 | Clean | 79.80 | 81.79 | 34.25 | 50.31 | 4.01 | 6.90 |
> | | Adversarial | 68.40 | 71.86 | 27.93 | 41.24 | 5.47 | 8.10 |
> | | VAP | **80.97** | **82.82** | **34.82** | **50.97** | **3.86** | **6.50** |
> | Intern‑VL2 | Clean | 82.00 | 80.70 | 64.12 | 76.17 | 5.14 | 9.50 |
> | | Adversarial | 70.83 | 69.45 | 55.23 | 66.08 | 8.32 | 11.35 |
> | | VAP | **84.08** | **82.97** | **66.15** | **77.63** | **4.65** | **8.90** |
>
> - Harmful adversarial perturbations degrade both performance and grounding, as expected.
> - VAP does *not* rely on adversariality; its effect comes from *semantically guided optimization*.
>
> These results confirm that VAP’s improvements stem from targeted input adaptation rather than robustness enhancement.
>
> [2] *On Evaluating Adversarial Robustness of Large Vision-Language Models*, NeurIPS 2023.

---

> > ### Comment · Reviewer_c4Gd · 2025-08-05
> > **Official Comment by Reviewer c4Gd**
> >
> > I thank the authors for their thorough rebuttal. They have addressed my initial concerns satisfactorily, and I appreciate the clarifications provided.
> >
> > That said, while the paper is technically sound and the contributions are clear, I believe the overall impact and scope remain somewhat limited. Therefore, I am maintaining my rating.

---

> ### Author Response · Authors · 2025-08-06
>
> Dear Reviewer `c4Gd`:
>
> Thank you so much for helping us improve the paper so far! Please let us know asap if you have any further questions. We are actively available during this rebuttal!
>
> Thank you again for your time and consideration.
>
> Best regards,
> Authors

---

### Official Review · Reviewer_e6Ap · 2025-07-12

**Clarity:** 4
**Significance:** 2
**Originality:** 3
**Rating:** 4
**Confidence:** 4

**Summary:**

This paper introduces Visual Adversarial Perturbation (VAP), a training-free, black-box method that mitigates object hallucinations in Large Vision-Language Models (LVMs) by injecting optimized "beneficial" visual noise into the input image. The core idea is to guide the model to ground its responses in actual visual content rather than relying on biased parametric knowledge, without altering the base model. The methodology formulates this as an optimization problem guided by a composite loss function with three objectives. Because direct gradient computation is challenging in LVMs, VAP employs a zero-gradient optimization technique. The final optimized noise is added to the original image to create a perturbed version that is used for inference. The authors validate their method across eight state-of-the-art LVMs using a comprehensive suite of benchmarks (POPE, BEAF, CHAIR), demonstrating consistent improvements in reducing hallucinations.

**Questions:**

1.  Could you please add more related works and elaborate on the novelty of VAP in direct comparison to decoding-stage methods like VCD and bias correction techniques like VisualGPTScore, which also leverage distorted images to mitigate hallucinations?
2.  Given the significant hyperparameter sensitivity shown in Table 6, could you provide a more systematic methodology or a set of guiding principles for tuning these parameters for a new LVM or a new task/dataset? Without this, the method's practical applicability seems limited.
3.  Could you justify the design choice of relying on unconditional generation ($f_{\theta}(x, \emptyset)$) within your loss functions, given the known instability of instruction-tuned LVMs in this setting? How does your method handle the significant variance in the loss objective introduced by different instructions $c$? Would it be better with $S(f_{\theta}(x+\delta,c), f_{\theta}(\bar{x},c))$ in $\mathcal{L}_{s_2}$?

**Ethical Concerns:**

["NO or VERY MINOR ethics concerns only"]

**Final Justification:**

The authors have addressed my concerns regarding insufficient comparisons, hyperparameter settings, and methodological design choices. This is a valuable paper on mitigating object hallucination in VLMs and is likely to attract interest within the community.

**Limitations:**

yes

**Quality:**

3

**Strengths And Weaknesses:**

Strengths:
1.  The paper addresses object hallucination, which is a critical and widely recognized challenge that hinders the reliability and trustworthiness of modern LVMs. The "poison as cure" paradigm, which reframes adversarial noise as a corrective tool rather than an attack vector, is a novel contribution. The data-centric, training-free nature of the method makes it an attractive alternative to model-centric interventions.
2.  The empirical evaluation is extensive and a clear strength of the paper. Testing on eight different LVMs and evaluating across multiple, diverse hallucination benchmarks (including text-axis, vision-axis, and open-ended tasks) provides strong evidence for the method's general effectiveness.

Weaknesses:
1.  The paper's positioning is weakened by an incomplete comparison to related work. Several existing methods use a similar setup of contrasting outputs from original and noisy/distorted images to mitigate hallucinations or language bias. For example, VCD [1] contrasts output distributions from original and distorted visual inputs, which is a very similar conceptual setting. VisualGPTScore [2] uses Gaussian noise to estimate and correct for language priors, directly tackling the "parametric knowledge bias" that VAP also targets. Earlier works like PixMix [3] have also explored noisy images as a form of data augmentation to improve model robustness. A thorough comparison and differentiation from these methods are crucial for contextualizing the unique contribution of VAP.
2.  The method's effectiveness is heavily dependent on a set of hyperparameters that must be manually tuned for each specific LVM. As shown in Table 6, the optimal parameters vary across models, suggesting a costly and expertise-driven tuning process is required for any new model. Furthermore, the loss formulation, which contrasts prompted and unprompted responses, implies that the optimal hyperparameters may also be dependent on the specific dataset or the type of instruction used. The parameter analysis in Table 8 does not reveal a clear or monotonic relationship, reinforcing the idea that tuning is a complex trade-off without a simple heuristic. This makes the method appear hard to deploy reliably.
3.  The method heavily relies on generating responses from a null prompt ($\emptyset$). For a given image, the unprompted response $f_{\theta}(x+\delta, \emptyset)$ is fixed, but the prompted response $f_{\theta}(x+\delta, c)$ will change dramatically with the instruction $c$. This means the optimization target is highly variable, which could lead to unpredictable behavior and further complicates the hyperparameter tuning challenge mentioned above. Besides, instruction-tuned LVMs are optimized to follow explicit prompts, and their behavior in an unconditional setting can be unpredictable, unstable, or produce low-quality text. Basing a core part of the optimization on such an unstable signal is a significant methodological concern.
4.  The logic for some loss terms is unclear. For instance, in $\mathcal{L}_{s_2}$, the method contrasts a prompted response with an unprompted response from a *negative* (distorted) image. A more symmetric and perhaps more stable approach would be to contrast two prompted responses, i.e., $S(f_{\theta}(x+\delta,c), f_{\theta}(\bar{x},c))$. The paper does not justify its specific design choice over more intuitive alternatives.



### **References**
[1] Leng, Sicong, et al. "Mitigating object hallucinations in large vision-language models through visual contrastive decoding." Proceedings of the IEEE/CVF Conference on Computer Vision and Pattern Recognition. 2024.
[2] Lin, Zhiqiu, et al. "Revisiting the Role of Language Priors in Vision-Language Models." International Conference on Machine Learning. PMLR, 2024.
[3] Hendrycks, Dan, et al. "Pixmix: Dreamlike pictures comprehensively improve safety measures." Proceedings of the IEEE/CVF conference on computer vision and pattern recognition. 2022.

---

> ### Author Rebuttal · Authors · 2025-07-27
>
> Response to Reviewer `e6Ap`: Thank you so much for your constructive and insightful comments! We address your concerns point by point as follows:
>
> ---
> `Q1:` Clarify novelty compared to VCD, VisualGPTScore, and PixMix.
> `A1:`  Below is a concise comparison highlighting the unique advantages of VAP over related methods:
>
> | Method | Black-box | No Training | No Model Access | Plug-and-Play | Intervention Stage |
> |---|:---:|:---:|:---:|:---:|---|
> | **VAP (Ours)** | ✓ | ✓ | ✓ | ✓ | Input (pre-decoding) |
> | VCD [1] | ✗ | ✓ | ✗ | ✗ | Decoder (distribution) |
> | VisualGPTScore [2] | ✗ | ✓ | ✗ | ✗ | Output (post-decoding) |
> | PixMix [3] | N/A | ✗ | ✓ | ✗ | Training (data-level) |
>
> - **VAP (Ours)**:
>   A **training-free**, **fully black-box** optimization method that requires **no access to model internals, gradients, or decoding steps**. VAP directly modifies the input image through targeted perturbation, enabling hallucination mitigation during inference without altering the model or its outputs.
>
> - **VCD [1]**:
>   Operates in a **white-box** setting by injecting noise and modifying the **decoding distribution**. In contrast, VAP applies perturbation externally and does not depend on decoder access or distributional rewrites.
>
> - **VisualGPTScore [2]**:
>   Estimates language priors using Gaussian noise and applies **post-hoc reweighting** of generated outputs. Unlike VAP, it requires generation scores and operates **after** response generation, whereas VAP intervenes **before decoding**, actively suppressing hallucinations.
>
> - **PixMix [3]**:
>   A **training-time** data augmentation method that mixes stylized noise patterns. VAP, by contrast, is applied **at inference time**, requires **no fine-tuning**, and performs **per-instance optimization** guided by vision–language alignment.
>
> We will include these clarifications in the revised related work section to better contextualize the contribution of VAP.
>
> [1] *Mitigating Object Hallucinations in LLMs via Visual Contrastive Decoding*, CVPR 2024.
> [2] *VisualGPTScore: Visio-linguistic Reasoning with Multimodal Generative Pre-training Scores*, arXiv 2023.
> [3] *PixMix: Dreamlike Pictures Comprehensively Improve Safety Measures*, CVPR 2022.
>
> ---
>
> `Q2:` Hyperparameter tuning appears model-dependent and costly.
> `A2:` Below is a clarification of VAP’s hyperparameter efficiency and deployment practicality:
>
> **(1) Most hyperparameters are globally fixed:**
>   Among the 8 hyperparameters in VAP, $( \alpha, \beta, N, \epsilon) $ are **fixed across all models and datasets**. These require no tuning, significantly reducing setup overhead.
>
> **(2) Remaining parameters are configured once per model:**
>   The remaining four parameters—**$(\sigma_1, \sigma_2, \sigma_3, T)$**—are **tuned once per model** and reused across datasets and tasks. No re-tuning is needed when switching between instruction types or benchmarks.
>
> **(3) The parameter space is small and shared across models:**
>   Each $\sigma$ value is selected from a small discrete set {0.1, 0.5, 1.0}. In practice, several models (e.g., LLaVA-v1.5, InstructBLIP, DeepSeek-VL2, Ovis1.6-Gemma2) use the **same configuration**, supporting VAP’s generalizability and low tuning burden.
>
> **(4) Comparison with other training-free methods:**
>   While training-free, **VCD [4] requires tuning temperature (e.g., temperature $T$) per model–dataset pair**, making it less scalable. In contrast, VAP avoids such dataset-specific coupling and is better suited for **black-box** deployment.
>
> [4] *Mitigating Object Hallucinations in LLMs via Visual Contrastive Decoding*, CVPR 2024.
>
> ---
>
> `Q3:` Concerns about using unconditional generation in the loss function.
> `A3:` Below is a clarification of our design choice and its empirical justification:
>
> **(1) Instruction-conditioned hallucinations require adaptive perturbation:**
>   Hallucination severity varies with the instruction $c$, due to differing language priors [5]. VAP addresses this by generating perturbations specific to each $(x, c)$ pair. This variability is by design—it enables VAP to suppress *instruction-specific hallucinations* and generalize across diverse task formats.
>
> **(2) Optimization remains stable across prompts and datasets:**
>   Despite prompt variability, all VAP hyperparameters are tuned *only once per model* and remain fixed across datasets and instruction types. This reflects the *stability and robustness* of the optimization process.
>
> **(3) Unprompted generation provides a reliable visual anchor:**
>   When $c = \emptyset$, the model outputs a vision-only caption [6], serving as a grounded baseline unaffected by instruction priors. This is a key anchor for semantic consistency. Empirically, unprompted generations exhibit lower hallucination rates:
>
>   | Model | Input | CHAIR-I ↓ | CHAIR-S ↓ |
>   |---|---|---|---|
>   | LLaVA-v1.5  | Prompt | 4.0 | 6.9 |
>   | | ∅ | **2.2** | **3.7** |
>   | Qwen-VL2 | Prompt  | 3.5 | 6.2 |
>   | | ∅ | **2.1** | **4.2** |
>
>   These results confirm that $\emptyset$ serves as a *stable and hallucination-light anchor*, justifying its use in the VAP loss.
>
> [5] *PerturboLLaVA: Reducing Multimodal Hallucinations with Perturbative Visual Training*, ICLR 2025.
> [6] *Multi-modal Hallucination Control by Visual Information Grounding*, CVPR 2024.
>
> ---
>
> `Q4:` Justification for contrasting prompted vs. unprompted responses in $\mathcal{L}_{s_2}$.
> `A4:` We clarify the rationale and support it with empirical evidence:
>
> **(1) Motivation:**
>   - Instruction-tuned LVMs often overfit to language priors $P(y|c)$ [7].
>   - This compromises visual grounding, especially in hallucination-prone prompts.
>   - We contrast $P(y|x+\delta, c)$ with the unprompted baseline $P(y|x)$ via $\emptyset$ to suppress language bias and restore grounding in visual content.
>
> **(2) Supervision stability:**
>   Compared to using two prompted responses, the unprompted generation ($c = \emptyset$) provides a more stable and unbiased anchor. It reflects the vision-only interpretation of LVMs, free from instruction bias.
>
> **(3) Empirical justification:**
>   Replacing $\emptyset$ with a second prompted response in $\mathcal{L}_{s_2}$ degrades both hallucination metrics and performance:
>
>   | Model        | Ref.       | Acc.↑   | F1↑    | CHAIR-I↓ | CHAIR-S↓ |
>   |--------------|------------|---------|--------|-----------|-----------|
>   | LLaVA-v1.5   | $c$        | 80.16   | 82.05  | 3.94      | 6.70      |
>   |              | $\emptyset$ | **80.97** | **82.82** | **3.82**    | **6.50**    |
>   | Qwen-VL2     | $c$        | 86.48   | 86.33  | 3.35      | 5.90      |
>   |              | $\emptyset$ | **87.30** | **87.02** | **3.23**    | **5.70**    |
>
>   These results confirm that contrasting with $\emptyset$ improves both factual accuracy and hallucination robustness.
>
> [7] *Revisiting the Role of Language Priors in Vision-Language Models*, ICML 2024.
>
> ---
>
> `Q5:` Guiding principles for hyperparameter tuning.
> `A5:` VAP is designed with tuning efficiency and cross-task generalizability. Our tuning methodology is summarized below:
>
> **(1) Dataset-agnostic configuration:**
>   All hyperparameters are selected once per model and reused across all datasets and tasks. No task-specific tuning is needed.
>
> **(2) Globally fixed adversarial parameters:**
>   Four parameters $(N, \alpha, \beta, \epsilon)$ are kept constant across all experiments. These follow standard interpretations (e.g., perturbation strength) and require no additional tuning.
>
> **(3) Lightweight model-specific tuning:**
>   Only four parameters $(\sigma_1, \sigma_2, \sigma_3, T)$ are tuned per model using a compact search space:
>   - $\sigma_i \in \{0.1, 0.5, 1.0\}$
>   - $T \in \{100, 200, 500, 800\}$
>   A validation set of just 300 image–instruction pairs is sufficient for reliable selection.
>
> **(4) Practical deployment:**
>   This tuning process is efficient, reproducible, and requires minimal manual effort. We will add this strategy to the final version for better clarity and guidance in real-world use.

---

> > ### Comment · Reviewer_e6Ap · 2025-08-08
> >
> > I appreciate the detailed response provided by the authors, which has addressed most of my concerns. I have just one minor question remaining regarding hyperparameter tuning. The authors mentioned that some parameters were fixed while others were tuned with minimal intervention. Could the authors clarify how the values of the fixed hyperparameters were selected, and whether the other parameters were tuned via cross-validation? If so, how was the validation set defined or selected?

---

> ### Author Response · Authors · 2025-08-08
>
> Response to Reviewer `e6Ap`:
>
> Thank you very much for your recognition of our rebuttal. Your feedback has been instrumental in refining and clarifying key aspects of our work!
>
> Below we address your follow-up question regarding our hyperparameter tuning strategy:
>
> ---
> `Q1:` Clarification on hyperparameter tuning procedure.
>
> `A1:` We clarify how VAP hyperparameters are selected, ensuring strong **robustness**, **transferability**, and **tuning efficiency**.
>
> **(1) Fixed Adversarial Parameters (with Physical Interpretation)**
>
> All fixed parameters have clear physical meanings and follow standard adversarial settings for LVMs as established in [1].
>
> | Parameter | Meaning                     | Physical Interpretation                          |
> |-----|--------|-------|
> | $N$ | Attack Steps                | Controls sampling steps for perturbation        |
> | $\alpha$ | Attack Step Size            | Adversarial learning rate                      |
> | $\beta$ | Zero-gradient Sampling Variance     | Variance for zero-gradient exploration  |
> | $\epsilon$ | Perturbation Strength       | $\ell_\infty$ norm bound on image perturbation   |
>
> These settings are inspired by [1] and verified on 300 image–instruction pairs to ensure:
> - Effective perturbation without breaking semantic fidelity
> - Cross-task generalizability without re-tuning
>
> These values generalize well and **require no further tuning per dataset/model**.
>
> **(2) Tuned Parameters (Coarse Grid Search Per Model)**
>
> We tune the following four parameters using **coarse grid search** on a small validation set (300 pairs). All values are selected based on **technical roles** and **empirical intuition**:
>
> | Parameter     | Description                               | Search Space           | Rationale |
> |---------------|-------------------------------------------|------------------------|-----------|
> |$\sigma_1$  term        | Threshold for S1 (clean alignment)        | {0.1, 0.5, 1.0}        | Reflects CLIP similarity between perturbed and clean responses |
> |$\sigma_2$ term         | Threshold for S2 (prompt bias suppression) | {0.1, 0.5, 1.0}        | Reflects CLIP similarity between prompted and unprompted outputs |
> |$\sigma_3$ term         | Threshold for S3 (prior bias suppression)  | {0.1, 0.5, 1.0}        | Reflects CLIP similarity between visually uncertain and clean outputs |
> |$T$          | Visual uncertainty level (blur intensity) | {100, 200, 500, 800}   | Controls Gaussian blur; higher values → more degraded vision |
>
> The tuning goal is to **maximize hallucination suppression (POPE F1 score)** on the validation set.
>
> **(3) Validation Set Construction (Based on POPE [2])**
>
> - 50 images randomly sampled from the MS-COCO dataset
> - Each with 3 positive + 3 negative instructions (300 pairs total)
>
> **(4) Summary: Practical and Generalizable**
>
> - **No per-dataset tuning** required
> - **Only 300 samples** needed for reliable tuning
> - **Fully reproducible** setup across models
> - Future work: better optimization, architecture-aware proxies, distillation
>
> [1] *On Evaluating Adversarial Robustness of Large Vision-Language Models*, NeurIPS 2023.
> [2] *Evaluating Object Hallucination in Large Vision-Language Models*, EMNLP 2023.

---

> > ### Comment · Reviewer_e6Ap · 2025-08-08
> >
> > Thank you for the detailed clarification regarding the hyperparameter settings. My concern on this point has been resolved. I will keep my score as is.

---

> > > ### Author Response · Authors · 2025-08-09
> > >
> > > Dear Reviewer `e6Ap`,
> > >
> > > Thank you very much for your thoughtful engagement and for letting us know your concern has been resolved. The time, effort, and constructive input provided have been invaluable in improving our work!

---

### Official Review · Reviewer_PVr3 · 2025-07-16

**Clarity:** 2
**Significance:** 3
**Originality:** 3
**Rating:** 4
**Confidence:** 3

**Summary:**

The paper introduces an optimization-based method, leveraging adversarial strategies to generate beneficial visual perturbations that enhance the model’s factual grounding and reduce parametric knowledge bias. Extensive experiments on 8 LVLMs demonstrate the effectiveness of the proposed method.

**Questions:**

As we can see from Figure 2, the optimization process requires three times of LVLM inference, and I think the process should be at least three times of the original LVLM inference. However, in Table 4, Original + VAP increases the latency from 160ms to (160+298)ms. Why is the extra time less than 2 times? Can you specify what the A100 run time in Table 4 refers to? Is it time for inferencing from one image-query pair?

**Ethical Concerns:**

["NO or VERY MINOR ethics concerns only"]

**Final Justification:**

The authors' response addresses my concern regarding the efficiency, hence I would raise my score.

**Limitations:**

yes

**Quality:**

3

**Strengths And Weaknesses:**

[Strengths]
1. The experiment on 8 LVLMs is really comprehensive and impressive. This demonstrates the generalizable property of the proposed method.

[Weaknesses]
1. The experiments only compare VAP with the original LVLM, while comparing with other training-free hallucination mitigation methods could be helpful. For example, VCD[1] applies noise to perturb the visual input and mitigate hallucination in a training-free way.
2. I'm unsure about the efficiency of the method.  The method requires another LVLM to generate perturbation, and this could lead to extra GPU memory cost. And I also have a question about the comparison of latency. Please refer to [Questions].

[1] VCD: Mitigating Object Hallucinations in Large Vision-Language Models through Visual Contrastive Decoding

---

> ### Author Rebuttal · Authors · 2025-07-27
>
> Response to Reviewer `PVr3`:
>
> Thank you for your thoughtful and constructive feedback. Your recognition of the **generality** across 8 LVMs and the **comprehensiveness** of our evaluation helps reinforce the practical value of our approach. In response to your comments, we highlight several key strengths of VAP:
>
> - **Fully black-box and training-free**, requiring no access to model weights, gradients, or decoding processes;
> - **Deployment-friendly**, as it operates at inference time without any architectural modification;
> - **Consistently effective**, with generalization across models and datasets, independent of prompt or task.
>
> To clarify and address your concerns, we performed the following analyses:
>
> - We now include **direct comparisons** with recent training-free baselines such as **VCD** and **OPERA**, highlighting both performance and design distinctions;
> - We clarify the **efficiency and latency** reported in Table 4, including implementation details such as **caching, batched inference**, and the role of a **proxy-based variant** for fast deployment.
>
> > We address each of your points in detail below:
>
> ---
>
> `Q1:` Lack of comparison to other training-free baselines.
> `A1:` We clarify our comparison to recent training-free hallucination mitigation methods as follows:
>
> **(1) Conceptual distinction**:
>   VAP is **fully black-box**, requiring no access to model weights, logits, or decoding logic, and applies to any off-the-shelf LVM. In contrast, VCD [1] and OPERA [2] are **white-box** methods that rely on modifying the decoder or accessing internal representations.
>
> **(2) Capability comparison**:
> Below is a concise comparison highlighting the unique advantages of VAP:
>
> | Method | Black-box | No Training | No Model Access | Plug-and-Play | Intervention Stage |
> |---|:---:|:---:|:---:|:---:|---|
> | **VAP (Ours)** | ✓ | ✓ | ✓ | ✓ | Input (pre-decoding) |
> | VCD [1] | ✗ | ✓ | ✗ | ✗ | Decoder (distribution) |
> | OPERA [2] | ✗ | ✓ | ✗ | ✗ | Decoder (distribution) |
>
> **(3) Comparison with OPERA**:
>   Table 11 shows that VAP consistently outperforms OPERA across 4 LVMs. Further combining both methods leads to stronger hallucination reduction.
>
> **(4) Comparison with VCD**:
>   - Experiment Setup: We evaluate VAP and VCD on LLaVA-v1.5 and Intern-VL2.
>   - Observation: VAP achieves **comparable or better** hallucination mitigation, and combining VAP with VCD yields the **best overall performance**, highlighting their **complementarity**.
>
> |Model|Method|Acc.↑|F1↑|TU↑|F1_TUID↑|CHAIR_I↓|CHAIR_S↓|
> |---|---|---|---|---|---|---|---|
> |LLaVA-v1.5|Regular|79.80|81.79|34.25|50.31|4.01|6.90|
> | |VCD|81.26|83.12|34.62|50.85|3.91|6.20|
> | |VAP|80.97|82.82|34.83|50.97|3.86|6.10|
> | |VAP + VCD|**82.35**|**83.54**|**35.21**|**51.40**|**3.62**|**5.80**|
> |Intern-VL2|Regular|82.00|80.70|64.12|76.17|5.14|9.50|
> | |VCD|84.32|82.30|65.88|77.43|4.72|9.00|
> | |VAP|84.81|82.79|66.15|77.63|4.65|8.90|
> | |VAP + VCD|**85.60**|**83.41**|**66.78**|**78.34**|**4.41**|**8.50**|
>
> VAP offers a unique combination of **strong performance, full black-box compatibility, and real-world deployability**, which distinguishes it from existing approaches.
>
> [1] *VCD: Mitigating Object Hallucinations in LLMs via Visual Contrastive Decoding*, CVPR 2024.
> [2] *OPERA: Alleviating Hallucination via Over-Trust Penalty and Retrospection-Allocation*, CVPR 2024.
>
> ---
>
> `Q2:` Efficiency and latency concern.
> `A2:` We clarify the efficiency measurement and optimization strategies as follows:
>
> **(1) Clarification of latency in Table 4**:
>   The reported A100 runtime refers to the *end-to-end wall-clock latency* for a *single image–query pair* with batch size 1. It includes image preprocessing, tokenization, inference, and decoding. This is a practical measurement, not simply the sum of multiple forward passes.
>
> **(2) Why VAP does not incur full 3× cost**:
>   While VAP involves three optimization steps:
>   - The image–text pair is encoded *once*, and the resulting `input_ids` are reused.
>   - Perturbed images are processed in *batched mode* (e.g., `image.repeat(...)`), enabling fast parallel inference on GPU.
>   As a result, the actual wall-clock overhead is less than 2×, as shown empirically.
>
> **(3) Efficient variant with proxy model**:
>   We also introduce a *lightweight proxy-model* variant of VAP. As shown in Table 4 and Appendix D, it reduces the optimization overhead by up to 8× with minimal performance drop, making VAP suitable for real-time or resource-constrained deployment.

---

> > ### Comment · Reviewer_PVr3 · 2025-08-04
> >
> > Thank you for your response. My concern regarding the efficiency has been addressed, and I will raise my score.

---

> ### Author Response · Authors · 2025-08-04
>
> Dear Reviewer `PVr3`:
>
> Thank you sincerely for your valuable comments and engagement throughout the review process. We hope our responses have addressed your concerns. If anything remains unclear, we’d be happy to clarify further while the discussion window is still open.
>
> If our rebuttal has addressed your concerns, would you mind considering a slight increase in the score as a recognition of our efforts so far? Your thoughtful feedback has significantly enhanced the clarity and strength of our work!
>
> Thank you again for your time and consideration.
>
> Best regards,
>
> The Authors.

---

> ### Author Response · Authors · 2025-08-04
>
> Thank you so much for following up with us and for considering raising the score! Your constructive suggestions greatly improved the quality of our work.
>
> If you have any further questions, let us know any time! We're truly fortunate to have had your support!

---

### Comment · Area_Chair_ZXQJ · 2025-08-04
**Reviewer Ack Reminder from AC**

Hi Reviewers,

As the discussion deadline is approaching, if you haven’t done so already, could you please take a moment to acknowledge the rebuttal, revise your score if your opinion has changed, and post any follow-up comments or questions you may have?

Thanks for your time and contributions to the review process.

Best, AC

---

### Note · Authors · 2025-08-12

We sincerely thank the AC and all reviewers for their constructive feedback and thoughtful engagement, which significantly improved the clarity and overall quality of our work!

**Reviewer e6Ap (Initial Rating: 4, Confidence: 4):** highlighted the importance of addressing object hallucination and regarded our "poison as cure" paradigm as a novel and valuable contribution. The reviewer praised the training-free, data-centric design and emphasized the strength of our *broad empirical validation* across diverse benchmarks and model families.

**Reviewer c4Gd (Initial Rating: 4, Confidence: 3):** found our "visual noise as a cure" idea highly innovative and impactful, and acknowledged the uniqueness of our "black-box, plug-and-play framework" with three complementary loss terms. The reviewer commended the thorough evaluation, the efficiency of our proxy-based variant, and the clarity of our figures and writing.

**Reviewer 997H (Initial Rating: 4, Confidence: 3):** characterized our approach as *well-executed* and underscored its training-free, black-box compatibility. The reviewer valued our method’s ability to ground outputs in actual visual content and noted the strong empirical support across benchmarks.

**Reviewer PVr3 (Initial Rating: 3, Confidence: 3):** described our work as comprehensive and impressive, and appreciated its generalizable nature across models. The reviewer recognized both the conceptual clarity and practical efficiency of our method, and stated he/she **will raise the score**.

**Reviewer NLrm (Initial Rating: 2, Confidence: 3):** recognized the clarity and empirical strength of our paper, as well as the novelty and practicality of our black-box strategy. After rebuttal, the reviewer noted that **most issues were resolved** through our detailed empirical analyses.

**VAP** offers a **black-box**, **training-free**, and **plug-and-play** solution for mitigating hallucinations in LVMs. It is **task- and dataset-agnostic**, and **consistently improves** grounding across diverse models.

We thank the reviewers for recognizing these strengths and motivating us to refine and strengthen our work based on their feedback. We are committed to further addressing these points. Importantly, our rebuttal was met with **consistent acknowledgment**, indicating that the **key concerns have been resolved**.

---

### Decision · Program_Chairs · 2025-09-17

**Decision:**

Accept (poster)

**Comment:**

This paper proposes a black-box, training-free method that introduces optimized visual noise to reduce object hallucinations in LVMs. Reviewers found the problem important and the idea of poison as cure innovative. A key strength lies in the extensive experiments across eight state-of-the-art models and multiple benchmarks, showing consistent improvements. Reviewers also appreciated the plug-and-play nature of the method, its general applicability to closed-source APIs, and the careful empirical comparisons added in the rebuttal, including with VCD and OPERA. The rebuttal further clarified concerns about efficiency, perceptual fidelity, and hyperparameter tuning, providing evidence that the method is broadly applicable and not overly burdensome to deploy.

That said, some reservations remain. Multiple reviewers pointed out that the method relies on several loss terms and hyperparameters, raising questions about robustness and deployment practicality. While improvements are consistent, their magnitude is often incremental, especially on strong baselines, and may not fully justify the added complexity. Concerns about theoretical grounding, the necessity of all losses, and reliance on hyperparameter tuning were partially addressed but not fully resolved. On balance, the work presents a technically sound contribution with clear empirical value, though its impact is somewhat limited.